# Median bed-material sediment particle size across rivers in the contiguous U.S.

Guta Wakbulcho Abeshu[1], Hong-Yi Li[1*], Zhenduo Zhu[2*], Zeli Tan[3], L. Ruby Leung[3]

[1]Department of Civil & Environmental Engineering, University of Houston, Texas, 77204, USA

[2]Department of Civil, Structural and Environmental Engineering, University at Buffalo, the State University of New York, New York, 14260, USA

[3]Pacific Northwest National Laboratory, Washington, 99352, USA

*Correspondence to*: Hong-Yi Li (hongyili.jadison@gmail.com) and Zhenduo Zhu (zhenduoz@buffalo.edu)

**Abstract**

Bed-material sediment particle size data, particularly for the median sediment particle size (D50), are critical for understanding and modeling riverine sediment transport. However, sediment particle size observations are primarily available at individual sites. Large-scale modeling and assessment of riverine sediment transport are limited by the lack of continuous regional maps of bed-material sediment particle size. We hence present a map of D50 over the contiguous U.S. in a vector format that

corresponds to approximately 2.7 million river segments (i.e., flowlines) in the National Hydrography Dataset Plus (NHDPlus) dataset. We develop the map in four steps: 1) collect and process the observed D50 data from 2577 U.S. Geological Survey stations or U.S. Army Corps of Engineers sampling locations; 2) collocate these data with the NHDPlus flowlines based on their geographic locations, resulting in 1691 flowlines with collocated D50 values; 3) develop a predictive model using the eXtreme Gradient Boosting (XGBoost) machine learning method based on the observed D50 data and the corresponding

climate, hydrology, geology and other attributes retrieved from the NHDPlus dataset; 4) estimate the D50 values for flowlines without observations using the XGBoost predictive model. We expect this map to be useful for various purposes, such as research in large-scale river sediment transport using model- and data-driven approaches, teaching environmental and earth system sciences, planning and managing floodplain zones, etc. The map is available at http://doi.org/10.5281/zenodo.4921987 (Li, Abeshu et al., 2021).

 **1 Introduction**

Bed-material sediment particle size information is critical for understanding and modeling riverine sediment processes, including sediment erosion, entrainment, deposition, and transportation. Various sedimentology formulas have been proposed to quantify the sediment processes, with sediment particle size being a critical parameter in those formulas (Meyer-Peter and Müller, 1948; Einstein, 1950; Engelund and Hansen, 1967; Ackers and White, 1973; Van Rijn, 1984; Parker 1990; Garcia and

Parker, 1991; Wu et al., 2000; An et al., 2021). Moreover, sediment particle size is a critical factor in riverine dynamics of heavy metal (Unda-Calvo et al., 2019; Zhang et al., 2020), nutrients (Xia et al., 2017; Glaser et al., 2020), microplastic (Corcoran et al., 2019; He et al., 2020), and fish habitats and benthic lives (Dalu et al., 2020; Rieck and Sullivian, 2020).

The sediment transport modes can be classified into bed-material load and wash load (Garcia, 2008). The bed-material load consists of all sizes of particles existing in a river bed regardless of whether they are being transported along the bed (bedload)

or in suspension (suspended load). Wash load consists of very fine particles (diameter less than 0.062 mm) that are always in suspension in the water and rarely reside on the bed (Garcia, 2008). Wash load is usually controlled by only land surface processes (soil erosion in hillslopes and transport from hillslopes into rivers), but not much by riverine hydraulic conditions (Garcia, 2008). In this study, we focus on the bed-material sediment particle size data that are critical in applying sediment transport formulas to estimate bed-material load. For example, median bed-material sediment particle size (denoted as D50,

i.e., the size larger than 50% of sediment particles) is one of the most important parameters in the Engelund-Hansen equation (Engelund and Hansen, 1967).

Despite the importance of bed-material sediment particle size, such data has limited availability due to the expensive costs of measuring and analyzing such data. As one of the most data-rich countries in the world, the United States (U.S.) collects and disseminates the sediment particle size data mainly through two federal agencies: The U.S. Geological Survey (USGS) and

the U.S. Army Corps of Engineers (USACE). USGS manages the most gauges and distributes the river-related measurements on the U.S. rivers. As of April 2021, there are 424948 stations with field/laboratory samples in the USGS water quality portal, among which 1.2% (3644) include bed-material sediment particle data for rivers over the contiguous U.S., and 0.6% (2277) have complete percentiles of bed-material sediment particle data for computing D50.

Spatial approximation, i.e., interpolation or extrapolation, is a typical method to overcome data sparsity when there is no

universal relationship between the variable of interest (e.g., D50) and other extensively available information. In the case of sediment particle size, a simple spatial approximation should be conducted within the same river system, assuming similar geological and hydrological settings. Here we denote a river system as the whole river network discharging to the ocean (or inland lakes) via the same outlet. Such a simple spatial approximation is nevertheless not feasible in many river systems, where there are few or no measurement data to support meaningful interpolation and extrapolation. Several studies have reported

empirical relationships between bed-material sediment particle size with river channel characteristics (e.g., channel slope) and flow regimes (Niño, 2002; Zhang et al., 2017). Such relations are nonetheless site-specific and not universal enough to apply over various river systems.

An alternative approach is to establish complex correlations between sediment particle size and other data that are extensively available over the contiguous U.S. Such correlations can then be applied across the U.S. for predicting sediment particle size.

Conventional linear or nonlinear regression methods usually require good prior knowledge of the mechanisms controlling sediment size distribution, and thus are not suitable for use to establish complex correlations when understanding of factors that control sediment size is somewhat limited. Machine learning offers an effective way forward because of its ability to establish nonlinear, complex predictive models without the prerequisite of sufficient process-based knowledge (Afan et al., 2016).

Therefore, our objective is to develop a spatial map of D50 over the contiguous U.S. rivers by establishing a predictive model between D50 and other extensively available hydroclimatological and geological data using state-of-the-art machine learning techniques. In the following, we describe the data in Section 2, introduce the machine learning model development in Section 3, and present our results in Section 4. We also explain the limitations of our method in Section 5, potential usage of the D50 map in Section 6, and data availability in Section 7. We finally conclude with Section 8.

**2 Data**

**2.1 Bed-material sediment particle size observations**

The USGS sediment data are available to the public through the National Water Information System (NWIS) water quality data portal. There are 3644 USGS stations with at least one sample of bed-material sediment particle size, but only 2277 stations have complete data to allow meaningful computation of D50, as shown in Figure 1a. There are 1367 USGS stations

with incomplete percentiles of bed-material sediment particle data, which can be divided into three groups: 1) 1183 stations have no effective percentiles provided; 2) 147 stations have only percentiles above the 50th percentile; 3) 37 stations have only percentiles below the 50th percentile. Therefore, we neglect these data in further analysis.

The USACE sediment particle size data are available in a technical report by Gaines and Priestas (2016). Gaines and Priestas (2016) include the bed-material sediment particle size samples taken at 442 locations along the Mississippi River main stem

between Head of Passes, Louisiana and Grafton, Illinois. We exclude the locations without exact geographic coordinates and eventually yield 300 locations, as shown in Figure 1a. In total, we have 2577 locations with complete bed-material sediment particle size percentiles to allow for the D50 calculation. At each location, the sediment particle size might have been sampled more than once at different times, although almost half of the locations are sampled only once (see Figure 1b for the histogram and Figure S1a for the spatial map). For about 94% of these stations, the latest samples were taken after the 1970s (see Figure

1c for the histogram and Figure S1b for the spatial map). We calculated the coefficient of variation (CV) for the 760 stations that have at least 5 samples over time. For the rest of the stations, the number of samples is too small for meaningful calculation of CV. For most of these 760 stations, the CV values range between 0.3 and 1.2, with a median of approximately 0.6 (see Figure S2). The small CV values indicate the good stability of D50 (at the same location) over time.

We compute the D50 values from the measured sediment particle size distributions in three steps: 1) the cumulative sediment size distribution curve is drawn with log-2-transformed sediment size (in mm) following the concept of the Ψ scale (Parker and Andrews, 1985). 2) A linear interpolation is performed between the percentiles smaller and larger than the 50th percentile to obtain the D50 value. 3) For the stations with multiple sampling times, a representative D50 value is computed as the mean D50 value from all the sampling times. We take the mean as a representative D50 to simply account for possible uncertainties in sampling and measurement. Although the sampling and measurement procedures are carefully designed (Edwards & Glysson, 1999), it is practically impossible to avoid uncertainties in such sampling and measurement procedures. Thus, we believe a representative D50 can be better estimated by taking a mean. The D50 values calculated following this procedure are denoted as "observed D50 values" to differentiate them from the predicted D50 values using machine learning techniques described later. Figure 1d shows the histogram of the computed D50 values in the Ψ scale. About 75% of these D50 values are between 0.0625 mm and 2.0 mm. It is suggested that a river can be a sand-bed or gravel-bed river if the D50 value is below or above 2.0 mm (Garcia, 2008). The D50 values computed from the observed sediment particle size distributions thus mostly reflect sand-bed river conditions, while only approximately 25% are gravel-bed rivers.

One might wonder how the sites with observed D50 values are distributed between small and large streams (e.g., whether or not smaller streams have more observed D50 data than larger streams). We use stream order (Figure S3a, b) and upstream drainage area (Figure S3c, d) as the indicators of stream size and examine the distributions of flowline lengths (Figure S3a, c) and D50 samples (Figure S3b, d), respectively. The total flowline length increases with the stream size (i.e., stream order or drainage area), which is expected since overall larger rivers have longer lengths. Interestingly, the number of D50 stations follows a bell distribution except for the largest stream order or drainage area, which is primarily due to the USACE measurements on the lower Mississippi River (198 sample locations). Therefore, there is no clear indication that larger or smaller streams dominate the D50 data points.

## 2.2 Predictive variables

The predictive variables are retrieved from the NHDPlus database (McKay et al., 2012) and additional attributes for the NHDPlus catchments from the ScienceBase dataset (Wieczorek et al., 2018). ScienceBase is a comprehensive scientific data and information management platform hosted by USGS (sciencebase.gov). In the medium resolution NHDPlus, there are about 2.7 million stream segments (average length of 1.93 km, denoted as flowlines from now on). NHDPlus directly provides 138 attributes of flowlines, most of which are descriptive instead of quantitative. We select eight quantitative attributes relevant to the channel geometry and hydrology, such as upstream drainage area, channel bed slope, mean annual flow velocity, sinuosity, etc. ScienceBase provides additional attributes related to the NHDPlus watersheds (local drainage area corresponding to a single flowline) and associated upstream drainage areas in thirteen themes (Wieczorek et al., 2018). We select 68 hydroclimatological and geological attributes from ScienceBase, such as climate, hydrologic, topographic, soil, and geologic conditions. In total, 76 attributes are selected as potential predictive variables for input to the machine learning algorithm. We

provide a detailed list of these predictive variables in Supplementary Table S1 and four illustrative maps in Supplementary Figure S4.

We then establish the spatial correspondence between the observed D50 values and the 76 predictive variables. In NHDPlus, there are ~26000 USGS stations associated with a portion of the flowlines through the common identifiers. This common identifier is unique for every flowline, but several USGS stations may be located on the same flowline and have the same common identifier. We match the 2277 USGS stations that have observed D50 values with stations in NHDPlus. Some of the 2277 USGS stations are not included in NHDPlus, so we obtain 1530 matching stations. The 300 USACE sampling locations are collocated with the flowlines via their geographic coordinates. There are 12 flowlines with 2 sampling locations and 2 flowlines with 3 sampling locations. In those cases, we assign the average of the D50 values of these USGS stations to the flowline. The mean length of the 14 flowlines is 6.63 km. In such a length, only two or three sampling locations cannot capture the spatial variability in a meaningful way. Therefore, we simply calculate the average without making further assumptions. We further exclude a few flowlines with missing attribute values. Finally, we have a total of 1691 flowlines corresponding to the observed D50 values, as shown in Figure 2. As such, in each of these 1691 flowlines, we have established a good correspondence between the observed D50 values and the 76 predictive attributes.

## 3 Model Development

Among various machine learning methods, eXtreme Gradient Boosting (XGBoost) is a version of the gradient tree boosting algorithm known for its high efficiency and superior performance in recent years (Chen and Guestrin, 2016; Zheng et al., 2019; Fan et al., 2021). The relations between the input predictors (e.g., watershed characteristics) and D50 are too complex to be established with traditional linear regression or dimensionless analysis methods. Therefore, we adopt XGBoost to develop a predictive model with the Optuna optimization framework (Akiba et al., 2019) for tuning hyperparameters and the SHapley Additive exPlanations (SHAP) (Lundberg and Lee, 2017) for feature importance analysis and thus feature selection. We also consider the representativeness of input predictors in the feature selection. More details are explained as follows.

### 3.1 XGBoost: eXtreme Gradient Boosting

Tree boosting is a machine learning framework that combines weak learners to develop a strong learner, where the base learners are decision trees that are trained sequentially, with the latter focusing on mistakes made by the preceding one. Gradient boosting machines are a family of tree boosting techniques. One of the most recent offspring of gradient boosting techniques is the XGBoost, a scalable end-to-end tree boosting system (Chen & Guestrin, 2016). It has been successfully utilized across a wide array of applications, such as snowpack estimation (Zheng et al., 2019) and water storage change in a large lake (Fan et al., 2021). XGBoost dataset is represented as $D = \{(X_i, Y_i), i = 1, 2, \ldots, N\}$, where $X_i = [X_{i1}, X_{i2}, X_{i3}, \ldots, X_{ip}]$ is a row vector with input features with real value elements and $Y_i \in R$. The tree ensemble model employs $M$ additive functions to predict the output of interest as

$$\hat{Y}_i = \phi(X_i) = \sum_{m=1}^{M} f_m(X_i) , \qquad f_m \epsilon \text{ F} \qquad\qquad 1$$

where F is the space of regression trees. The model is trained in an additive manner by minimizing a regularized objective to learn the set of functions employed in the model. At each iteration, a differentiable convex loss function that measures the difference between the prediction $\hat{Y}_i$ and the target $Y_i$ is computed, and the model is also penalized for the complexity of the regression tree functions.

## 3.2 Tuning Hyperparameters

Tuning hyperparameters is a cumbersome task and is often performed by reducing the parameter search space through randomized search and applying a grid search on the reduced space. Alternatively, hyperparameter optimization frameworks like Hyperopt (Bergstra et al., 2015) and Optuna (Akiba et al., 2019) are commonly preferred since they can continually narrow down the bulky hyperparameter search space to an optimal space based on the preceding results. This study implements Optuna with a Tree-structured Parzen Estimator (TPE) parameter sampling framework to obtain the optimal hyperparameter sets.

The procedure for tuning hyperparameters relies on two major components: cross-validation and evaluation metrics. Cross-validation measures the model's predictive power with a given hyperparameter set by dividing a dataset into folds. In $k$-fold cross-validation, the dataset is randomly split into $k$ mutually exclusive subsets of approximately equal size as, $D = \{D_1, D_2, D_3, \ldots, D_k\}$. In each iteration, $k-1$ folds of $D$ are used for training, and the remaining one is used for validation. The predictions resulting from a given set of hyperparameters are made by iterating through the folds, so the model is trained and validated $k$ times. Hence, $k$ model performance values and the mean value is reported as the model performance for this set of hyperparameters. Optuna allows the use of user-defined metrics for model evaluation during the $k$-fold cross-validation. Taking advantage of this structure, we use the Kling-Gupta Efficiency (KGE) (Gupta et al., 2009) as the model performance metric.

$$KGE = 1 - \sqrt{(1-r)^2 + \left(1 - \frac{\sigma_{sim}}{\sigma_{obs}}\right)^2 + \left(1 - \frac{\mu_{sim}}{\mu_{obs}}\right)^2} \qquad\qquad 2$$

where $\sigma$ is the standard deviation, $\mu$ is the mean, and $r$ is the linear correlation between the observed and simulated series. A perfect agreement between observation and simulation gives the theoretical maximum KGE value at 1.0. The higher the KGE value, the closer the match between the observed and simulated series. KGE offers some advantages over commonly used metrics like root mean squared errors (RMSE) or the coefficient of determination ($R^2$) because 1) it is not dominated by relatively large values; and 2) it simultaneously captures both the magnitude and phase differences between the observed and simulated series (Gupta et al., 2009).

## 3.3 Feature Selection

Feature selection is also an essential step in developing a simpler model that is still capable of reasonably predicting the target variable with fewer attributes. Feature importance is a technique of computing each predictive variable's degree of contribution towards the optimal prediction model, which can be used for determining feature selection. The approaches of computing feature importance scores include correlation coefficient, the coefficients calculated as part of decision trees, or advanced approaches like SHAP (SHapley Additive exPlanations) (Lundberg and Lee, 2017). In this study, we use the mean absolute SHAP values as feature importance measures. Initially, we begin with 76 predictive variables. For feature selection purposes, we add a new "predictor" of randomly generated real number values. We train the model and compare the feature importance scores (i.e., the mean absolute SHAP values) of all predictors. Then, all attributes with scores less than the random number attribute are dropped out. The procedure is repeated using the new set of predictors until the random number attribute is the least important feature.

Then, we further examine the representativeness of the data by comparing the ranges of the selected features between the D50-available data and the nationwide data. We use the 2.5th and 97.5th percentiles to represent the lower and higher ends of ranges in the available data. We do not directly use the absolute min/max values to avoid the impacts of outliers. We then calculate the percentage of the nationwide data below the 2.5th percentile of the available data. A percentage value of no more than 10% indicates a good match of lower ends between the available and nationwide data. Similarly, we calculate the percentage of the nationwide data above the 97.5th percentile of the available data. A percentage value of no more than 10% indicates a good match of upper ends between the available and nationwide data. Taking together, for any feature, if more than 80% of the nationwide data are located within the 2.5th and 97.5th percentiles of the available data, we consider that the available data is sufficiently representative of the nationwide data for this specific feature. Otherwise, a feature is considered non-representative thus is removed from model development. Lastly, the remaining features are utilized for tuning the final optimal set of hyperparameter values.

## 3.4 General Steps

The general steps of the model development procedure are as follows and illustrated in Figure 3.

1. The predictors are scaled using the Minimum-Maximum scaler method, i.e., all features will be transformed into a range of [0,1]. The main advantage of having this bounded range normalization is that it can suppress the effect of outliers.
2. The dataset is randomly split into training (70%) and testing (30%) sets, respectively. Only the training data are used in steps 3 and 4, while the testing data are reserved for step 5.
3. Optuna and k-fold (k=5) cross-validation are used for tuning hyperparameters, with a maximum tree of 5000 and an early stopping value of 50. The objective function for the hyperparameter optimization procedure is to maximize the mean Kling-Gupta Efficiency (KGE) value returned from the k-fold cross-validation.

4. Feature selection is performed as described in section 3.3, so step 3 is repeated with the new and smaller set of predictors. Steps 3 and 4 are repeated until no more predictors can be excluded.

5. The final model is developed by fitting on the whole training data using the optimal hyperparameters, and evaluated using the testing data reserved in step 2.

6. The model from step 5 is used to predict the D50 values for the contiguous U.S. river flowlines.

## 4 Results

We discuss our results in three steps: the subset of flowlines as the basis to formulate our predictive model, the development and validation of our predictive model, and the national D50 map derived based on the predictive model.

### 4.1 Measured D50

Figure 2 shows the 1691 flowlines with the associated observed D50 values. The Mississippi River has relatively denser measurements attributed to the USACE database, while the southwest (e.g., the Rio Grande) and the Great Basin have fewer measurements. Overall, the 1691 flowlines are distributed throughout the contiguous United States, providing a good spatial representation of the NHDPlus flowlines. Similar to all observed D50 values in Figure 1b, most of the D50 values associated with the flowlines represent sand-bed rivers (D50 < 2.0 mm). Larger-D50 (> 2.0 mm) flowlines are mainly located in the basins of California, Upper Colorado, Missouri, Ohio and Upper Mississippi.

### 4.2 Predictive Model

#### 4.2.1 Feature Selection

After iterations of feature selection (procedure described in sections 3.3 and 3.4), 12 out of 76 predictive variables, or predictors, are eventually selected. Firstly, 13 variables are identified as more significant than a random-number input vector based on the mean absolute SHAP value, as shown in Table 1. 2 out of 8 channel characteristics and 11 out of 68 basin characteristics remain as the significant predictors (see Table 1 for description). The most important predictor is found to be the soil erodibility factor (Soil_erod_factor), followed by average annual wet day (Ann_wet_days) and mean annual snow as a percent of total precipitation (Ann_snow_perc).

These three basin-related predictors rank higher than the two channel-related characteristics. Channel slope (Slope) and distance between flowline and the river mouth (Chan_length) are found to be the most important channel characteristics for predicting D50, which agrees with the downstream fining phenomena and sediment transport mechanisms (Nino, 2002). It is somewhat surprising that some hydraulic channel characteristics such as mean annual flow velocity are not included in the final feature selection. Studies on river hydraulics show relations between channel flow (i.e., velocity and water depth) and channel bed characteristics (i.e., slope and roughness), such as the Manning's equation, Chezy's law, etc., and channel bed roughness can be related to bed sediment size (Garcia, 2008). However, the feature selection with the XGBoost model and

SHAP value indicates that mean annual flow velocity may not be a good predictor for D50 in this case. A possible reason is that mean annual flow velocity is dependent on some of the selected features such as Ann_wet_days, Slope, etc., so excluding this variable avoids overfitting the data.

It should be noted that the ranking of feature importance according to the mean absolute SHAP values is quite different from the correlation coefficients between D50 and predictors, as shown in Figure 4. Soil_erod_factor and Ann_wet_days, the two most important features in Table 1, have correlation coefficients of only 0.08 and 0.06, respectively. Ann_snow_perc has the strongest correlation with D50, with a correlation coefficient of 0.29. The individual scatter plots between D50 and each of the selected features do not show any apparent relationship between D50 and any single feature (see Supplementary Figure S5),

indicating that there might exist some higher-order interactions among the predictors which the traditional regression analysis cannot reveal.

We further examine the representativeness of the 1691 flowlines with observed D50 values of the rest flowlines included in the NHDPlus database in terms of the ranges of the 13 features. For convenience, we denote the subset of NHDPlus data associated with the 1691 flowlines as *D50-available*, and the whole NHDPlus database as *nationwide*. For most of the 13

features, the percentages of the nationwide data that are beyond the lower or higher ends of the D50-availableare no more than 10%, except for channel slope, i.e., "Channel_mean_slope". Table 2 also lists the relative difference in the 25th, 50th and 75th percentiles between the D50-available and nationwide data. For instance, for the 25th percentile, we calculated the relative difference as the ratio of the difference between the 25th percentile of the D50-available data and that of the nationwide data over the average between the 25th percentile of the D50-available data and that of the nationwide data. This relative difference

is less than 0.5 for most of the features, again except for "Channel_mean_slope". For a better visual illustration, Figure 5 shows the cumulative distribution functions (CDFs) and corresponding 5th-, 25th-, 50th-, 75th-, and 95th-percentiles. The CDFs are close between the D50-available and nationwide data, except for "Channel_mean_slope". A similar message can be seen from the box plots in Figure S6. In addition, we would like to point out that the 1691 sampling stations we use to train and test our model are located across the whole contiguous U.S., hence geographically representative. Therefore, we conclude that the

1691 flowlines with observed D50 values are representative enough of all the flowlines nationwide in terms of the 12 input predictors, except for "Channel_mean_slope".

We remove "Channel_mean_slope" and use the remaining 12 predictors to develop the final model, following the same model training and testing procedures as before. Figure 6 shows the comparison of model performances between the previous and new models. The model performance metrics are similar. Actually, $R^2$ became slightly better in both training (0.834 vs. 0.830)

and testing (0.405 vs. 0.367), while KGE became slightly worse in training (0.775 vs. 0.794) but better in testing (0.527 vs. 0.513). The slight decrease of KGE in training data is reasonable since the model hyperparameter tuning was based on the objective of maximizing KGE, and losing one predictor will slightly reduce the space of parameter tuning. Nevertheless, now the KGE value in the testing phase is closer to that in the training phase.

Although feature selection sheds light on the contribution of input variables to model outputs, a drawback of the machine

learning technique is that it cannot explain mechanistically why selected features are more important than unselected ones.

Therefore, the goal of feature selection is to find the best (i.e., most robust) input variables to feed the best model for D50 predictions. If a different machine learning algorithm from XGBoost is used, the selected features, especially their rankings, can be different. Feature selection is dependent on the selection of the algorithm, so the selected features in this study should not be directly used in other models or studies. In the 12 selected predictors, only one is directly related to the channel

processes. The remaining 11 are all land features, and their mechanistic connections with D50 are rather mysterious at this stage, which could be considered as empirical evidence on the likely causal, yet highly complicated relationships between D50 and the land features, and hopefully inspire future studies to shed light on the underlying mechanisms.

### 4.2.2 Model Hyperparameters and Performance

Table 3 shows the tuned hyperparameters of the best XGBoost model that is trained using the 12 selected predictors and 70%
of the training data set. For a detailed explanation of the hyperparameters please refer to Chen and Guestrin (2016). Figure 6 shows the performance of the optimal XGBoost model on the training and testing data sets, respectively. Here we consider an optimal model based on two criteria: 1) the model performance is satisfactory in both the training and testing phases, and indicated by good metrics values (e.g., KGE in this study), and 2) the model performance is relatively consistent between the training and testing phases. Here the optimal XGBoost model gives the KGE value 0.75 for training and 0.528 for testing. The
testing value is above 0.5, suggesting satisfactory model performance (Gupta et al., 2009; Knoben et al., 2019). The performance on the testing data is noticeably worse than that on the training data, as expected. This difference is nevertheless acceptable given the complexity of the prediction problem. The relatively consistent model performance between the training and testing phase suggests that the model validation (via the testing phase) is successful.

### 4.2.3 Model Sensitivity Analysis

We carry out further analysis to shed light on how the modeling results may be sensitive to some of the key steps as outlined in Section 3.4. We focus on Steps 2 to 4 only because Steps 1 and 5 are standard practice, and Step 6 utilizes the modeling results.

For Step 2, the 2/3 (train) and 1/3(test) split is typical in machine learning for splitting training and testing data. This can be
readjusted up to 4/5(train) and 1/5(test) if the total sample size is sufficiently large, which is nonetheless not the case here. For Step 3, we test the sensitivity on the choices of model performance metrics and k value, respectively. For the model performance metrics, we have also tried NSE and $R^2$ and found that using KGE gave better model performance (Figure S7) due to two reasons: 1) a much smaller percentage of bias (PBIAS) and 2) visually better alignment between the simulated and observed D50 values along the 1:1 line. The choice of k value is usually 5 or 10 depending on the training sample size. We
use 5 since using 10 significantly reduces the number of samples per fold, and the left-out sample will be too small for validation during cross-validation. Increasing k-fold to 6 or decreasing it to 4 still gives a similar satisfactory performance in both the training and testing phases, with training/testing KGE of 0.759/0.505 and 0.795/0.512, respectively. For Step 4, we

evaluate the model sensitivity to each selected feature by dropping one of the 12 variables at a time and repeating the same modeling procedure for the remaining 12 variables. Figure 7 shows that dropping the variables leads to the model performance dropping below KGE 0.5 during the testing phase in most cases. The change in testing phase KGE ranges between 4 – 22 %. The largest changes are observed when dropping R_factor and Mean_elev. Even with those two, the KGE difference between the training and testing phases increases from 0.28 to 0.36 by including them as predictors. Thus, all the variables remaining after feature selections play a significant role in the final model.

## 4.3 National Map

Using the developed machine learning model and NHDPlus channel/basin characteristics data, we are able to produce a national map of bed sediment D50 values (Figure 8). The spatial pattern of D50 in Figure 8 are generally consistent with the observed D50 in Figure 2. High D50 values are mostly distributed in the west coast, upper Missouri and Ohio regions, and low D50 values are concentrated in the east coast. The consistency between Figures 2 and 8 suggests that the observed D50 data are reasonably representative of the whole contiguous U.S., despite the sparse distribution. Given that the testing data set is independent of the training dataset, we expect that the error statistics derived for the testing data should be relatively consistent with the error statistics in applying the model to derive the national map of D50. To our best knowledge, it is the first-of-its-kind D50 data for the whole contiguous U.S. Such a D50 map is mostly valuable to support the parameterization of large-scale sediment modelling at the regional or national scale, which has been very challenging, if not impossible, before this map.

## 5 Limitations of the method

The predicted D50 values may be subject to several limitations despite using state-of-the-art machine learning techniques to develop the predictive model. These limitations include (1) Limited data availability. Although the 1691 observed D50 values are adequately representative of the contiguous U.S. (i.e., consistent spatial patterns between Figures 3 and 8), limited data availability prevents us from establishing a separate predictive model for each river basin. For example, there is little observed D50 data in the Rio Grande and Great Basin, so the predicted D50 values over these basins should be used cautiously. (2) Our methodology is statistical in nature and lacks explicit process-based understanding. For example, Figure 6 shows the model tends to overestimate D50 for smaller D50 values (particularly < 0.25 mm) and underestimate D50 for larger D50 values (particularly > 1 mm). However, in various trials we have performed, the current result is closest to the 1:1 relationship based on both the KGE metric and visual check. Further process-based understanding of this systematic bias is beyond the scope of this study because it would require a) a highly-integrated, process-based model that considers at least sediment erosion, deposition and transport processes in both hillslopes and channels, and b) well-designed numerical experiments to isolate the dominant processes and controlling factors. (3) We have not explicitly incorporated the effects of lakes and reservoirs but rather assumed these effects have been indirectly reflected in the NHDPlus hydrologic attributes adopted in the predictive model. (4) The bed sediment at the gage station may not always be representative of the reach. Edwards and Glysson (1999)

characterized how most of the bed sediment samples were collected and composited at a cross-section by the USGS over the
years. Gage stations are established at cross-sections in the stream where flow measurements are convenient and with
conditions conducive to high-quality flow measurements – the issue of whether the bed sediment composition represents the
reach is generally not taken into account when the gage station location is established. As such, our predictive results are
certainly not free from uncertainties. Therefore, we recommend using our D50 map for sediment modeling and assessment at
the regional or national scales instead of local studies at the individual river segment.

## 6 Potential usage

The D50 map might be used for large-scale sediment transport modeling over the whole contiguous U.S., or a major river
basin such as the Mississippi River basin. For example, we have tested the usage of the new D50 dataset within a large-scale
suspended sediment modeling framework (Li, Tan et al., 2021), and our successful model validation against the USGS
observed suspended sediment load over multiple stations suggests the good value of such a national scale D50 dataset. There
is inevitably some uncertainty embedded in this map sourced from the original D50 observations and NHDPlus attributes, the
XGBoost modeling, and the spatial extrapolation process. This uncertainty should be taken into account while utilizing this
map for regional-scale assessment or modeling.

## 7 Data availability

The national D50 map is freely available at http://doi.org/10.5281/zenodo.4921987 (Li, Abeshu et al., 2021). The input data
are obtained from the USGS water quality portal (https://nwis.waterdata.usgs.gov/usa/nwis/qwdata), NHDPlus
(https://www.epa.gov/waterdata/nhdplus-national-data ) and ScienceBase (https://doi.org/10.5066/F7765D7V).

## 8 Conclusions

We develop a new national map of the median bed sediment particle size by combining the USGS sediment observations, the
channel and watershed characteristics from NHDPlus and ScienceBase, and state-of-the-art machine learning techniques.
Despite the limitations, the map is highly valuable for sediment modeling and assessment at the regional and larger scales,
which has not been feasible previously.

## Author contributions

GA conducted the analysis. HL and ZZ designed the study. HL and RL conceived the idea. All the authors contributed to the
writing.

## Competing interests

The authors declare that there is no conflict of interest.

## Acknowledgments

This research is supported by the Office of Science of the U.S. Department of Energy as part of the Earth System Model
Development program area through the Energy Exascale Earth System Model (E3SM) project. The Pacific Northwest National
Laboratory is operated by Battelle for the U.S. Department of Energy under Contract DE-AC05-76RLO1830.

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

**Table 1. Most important predictors according to the feature selection**

| Predictor | | Description | Mean Absolute SHAP Value (w/ Slope) | Mean Absolute SHAP Value (w/o Slope) |
|---|---|---|---|---|
| Name used in this study | Name in NHDPlus | | | |
| Basin_slope | TOT_Basin_slope | Average topographic slope within the upstream drainage area | 0.30 | 0.42 |
| Ann_runoff | TOT_RUN | Average annual runoff within the upstream drainage area | 0.23 | 0.41 |
| Chan_length | Pathlength | Distance from the downstream end of a flowline to the end of the network (river mouth) | 0.34 | 0.39 |
| Ann_snow_perc | TOT_PRSNOW | Mean annual snow as a percent of total precipitation | 0.37 | 0.37 |
| Aridity_index | AI | Aridity index defined as the ratio of annual mean potential evaporation to annual mean precipitation | 0.29 | 0.37 |
| Ann_wet_days | TOT_WDANN | Average annual number of wet days | 0.46 | 0.36 |
| Mean_temp | TOT_WBM_TAV | Average mean annual temperature within the upstream drainage area | 0.29 | 0.35 |
| R_factor | TOT_RFACT | R factor of Universal Soil Loss Equation | 0.34 | 0.35 |
| T_Qsub | TOT_CONTACT | Time it takes for water to drain along subsurface flow paths to the stream | 0.31 | 0.34 |
| Mean_elev | TOT_ELEV_MEAN | Average surface elevation within the upstream drainage area | 0.29 | 0.31 |
| Qsat_to_Qtotal | TOT_SATOF | Annual saturation overland flow as a percent of total runoff | 0.33 | 0.27 |
| Soil_erod_factor | TOT_KFACT | Soil erodibility factor of Universal Soil Loss Equation | 0.51 | 0.22 |
| Channel_mean_slope | Slope | Channel Slope | 0.36 | |

Note: here we use the same names as those in the NHDPlus attribute tables, but moderately revise the description using terminologies that can be understood by a broader audience.

**Table 2: Comparison of the ranges and percentiles of 13 input features between the D50-available and nationwide datasets.**

| Attributes | Percent of nationwide data that | | Relative difference in percentiles between the D50-available and nationwide data | | |
|---|---|---|---|---|---|
| | below 2.5th of the D50-available data | above 97.5th of the D50-available data | 25th | 50th | 75th |
| Soil_erod_factor | 5.1 | 3.4 | 0.07 | 0.03 | 0.03 |
| R_factor | 6.9 | 10.2 | 0.22 | 0.09 | 0.44 |
| Ann_wet_days | 2.8 | 3.1 | 0.01 | 0.07 | 0.03 |
| Ann_snow_perc | 0.0 | 3.9 | 0.71 | 0.20 | 0.12 |
| Channel_mean_slope | 0.0 | **19.9** | 1.72 | 1.44 | 1.43 |
| Chan_length | 2.3 | 5.2 | 0.07 | 0.21 | 0.04 |
| Ann_runoff | 4.3 | 1.6 | 0.00 | 0.28 | 0.23 |
| Qsat_to_Qtotal | 0.0 | 7.1 | 2.00 | 0.13 | 0.38 |
| T_Qsub | 6.2 | 3.1 | 0.68 | 0.59 | 0.37 |
| Basin_slope | 9.3 | 4.8 | 0.17 | 0.04 | 0.10 |
| Mean_elev | 9.9 | 2.5 | 0.35 | 0.27 | 0.22 |
| Aridity_index | 2.6 | 6.5 | 0.07 | 0.14 | 0.03 |
| Mean_temp | 4.5 | 8.6 | 0.02 | 0.12 | 0.18 |



**Table 3. Optimal value of the XGBoost model hyperparameters**

| Hyperparameter | Optimal Value | Tuning Range |
|---|---|---|
| learning_rate | 0.442 | [0,1] |
| min_split_loss | 11 | [0,∞] |
| max_depth | 7 | [0,∞] |
| min_child_weight | 21 | [0,∞] |
| max_delta_step | 41 | [0,∞] |
| subsample | 0.408 | [0,1] |
| colsample_bytree | 0.741 | [0,1] |
| reg_lambda | 0.311 | [0,∞] |
| reg_alpha | 4.054 | [0,∞] |
| n_estimators | 178 | [1,∞] |


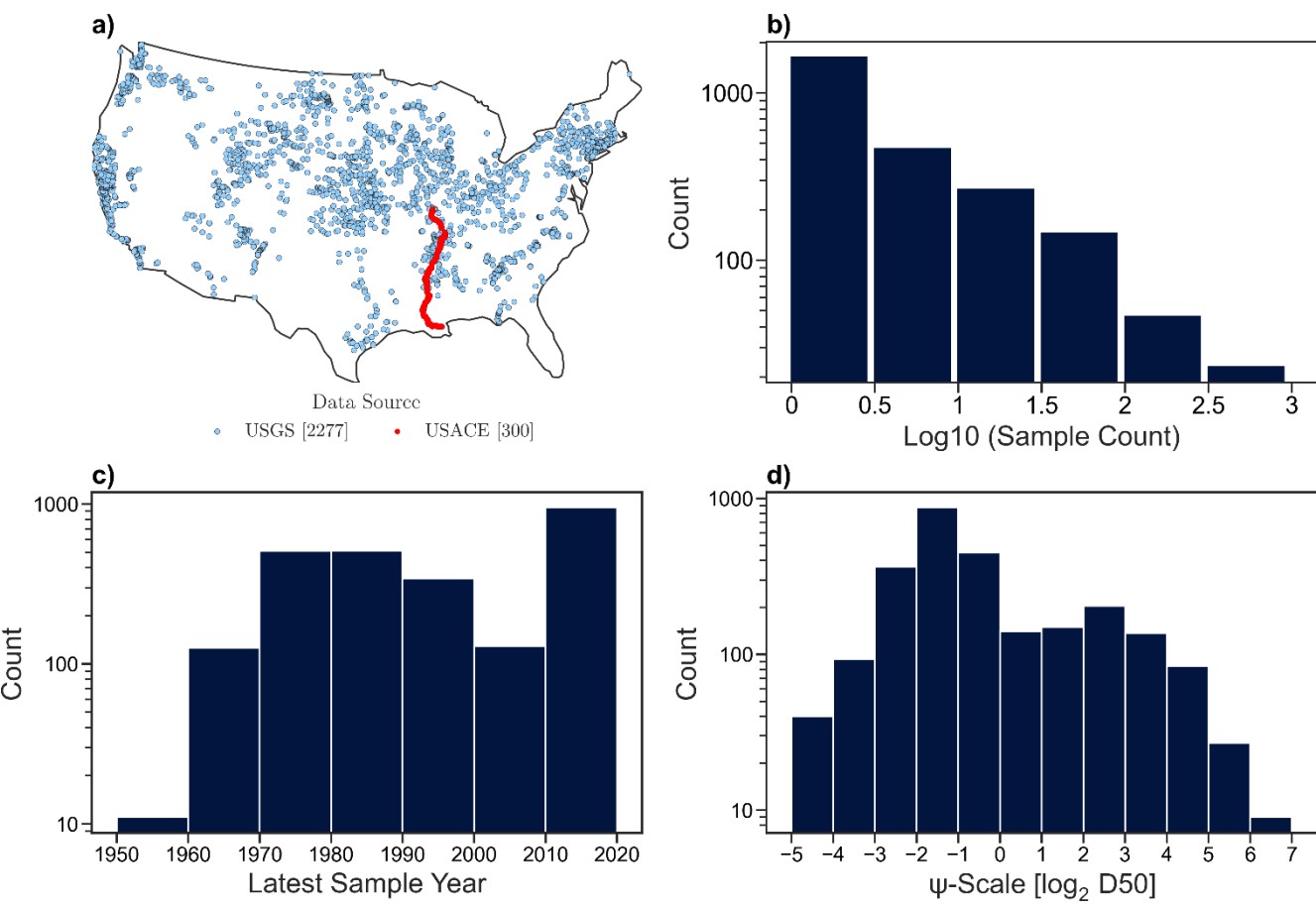


**Figure 1: Sediment sample stations. a. Locations of 2277 USGS stations and 300 USACE sampling locations; b. Number of samples at each station/location; c. Histogram of latest sample years; d. Histogram of D50 values in the Ψ scale (log₂D50, D50 in mm).**


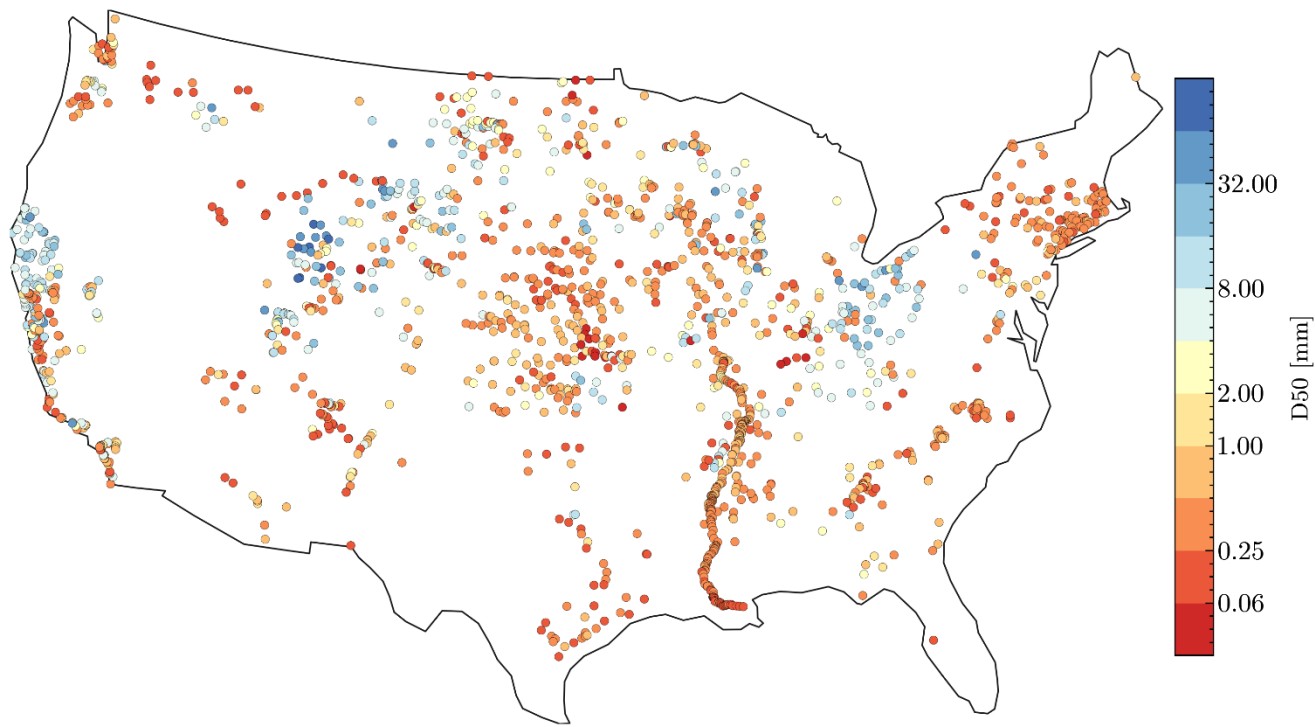

**Figure 2. 1691 flowlines with measured D50.**



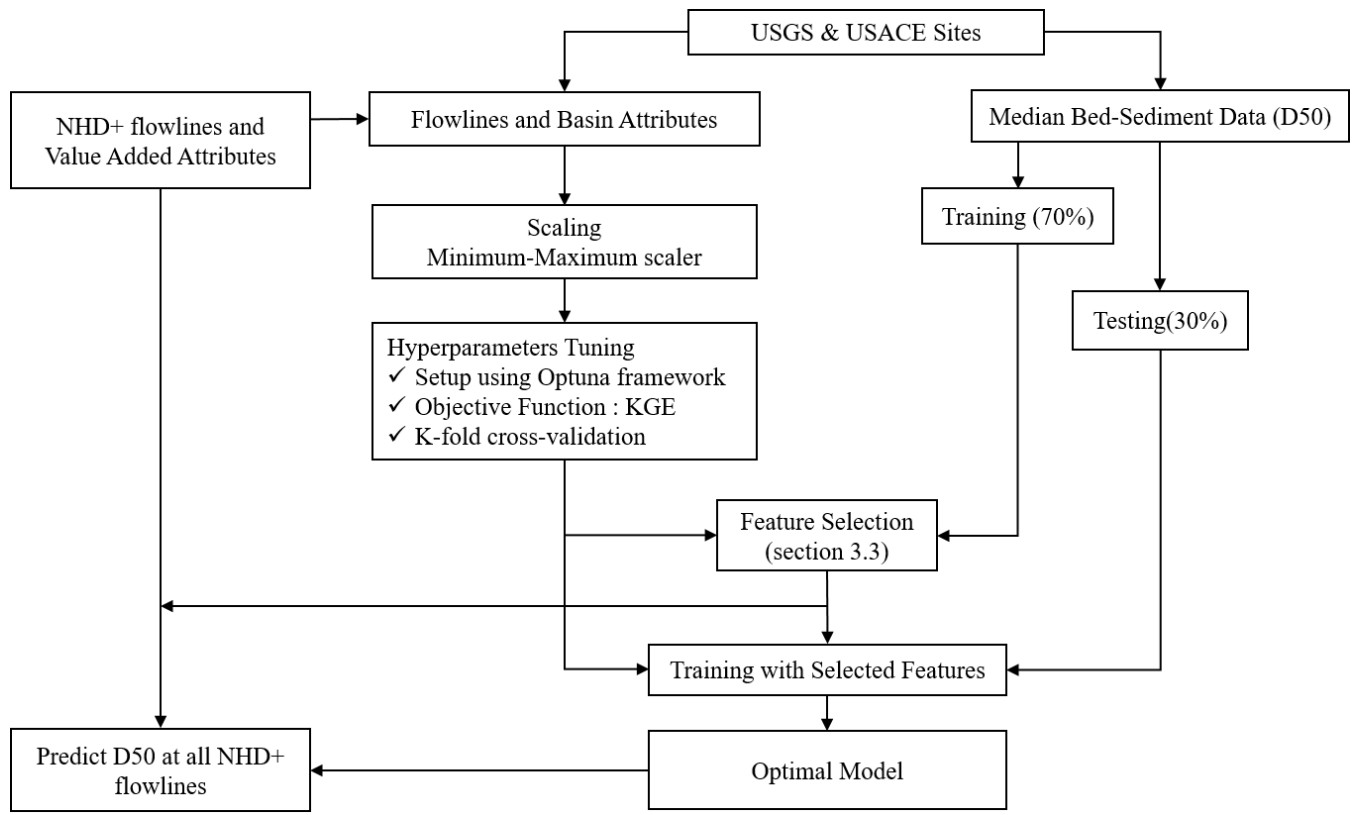

**Figure 3: Flowchart for XGBoost training and prediction.**


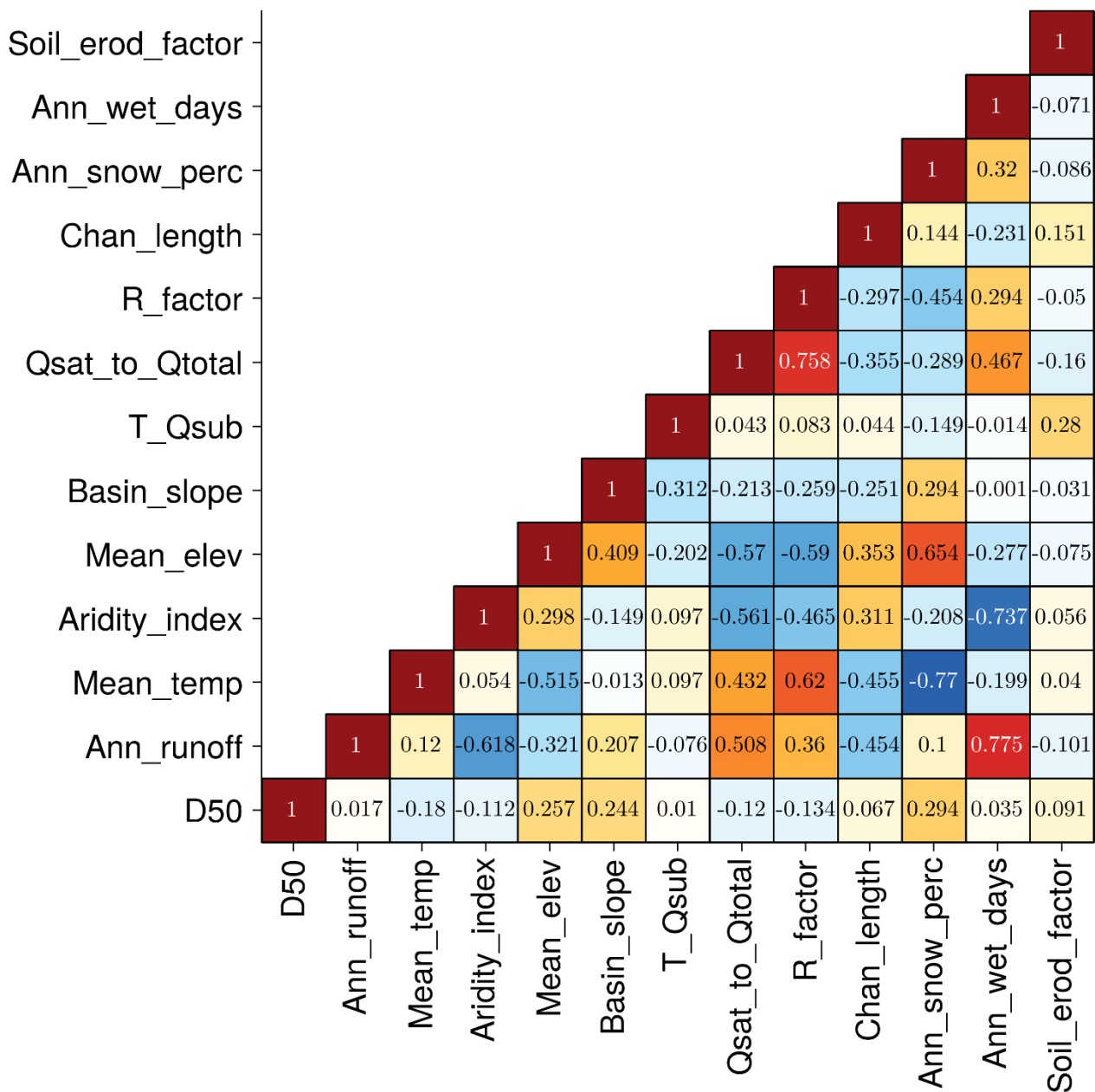

**Figure 4. Correlation coefficients among D50 and the 12 selected predictors.**

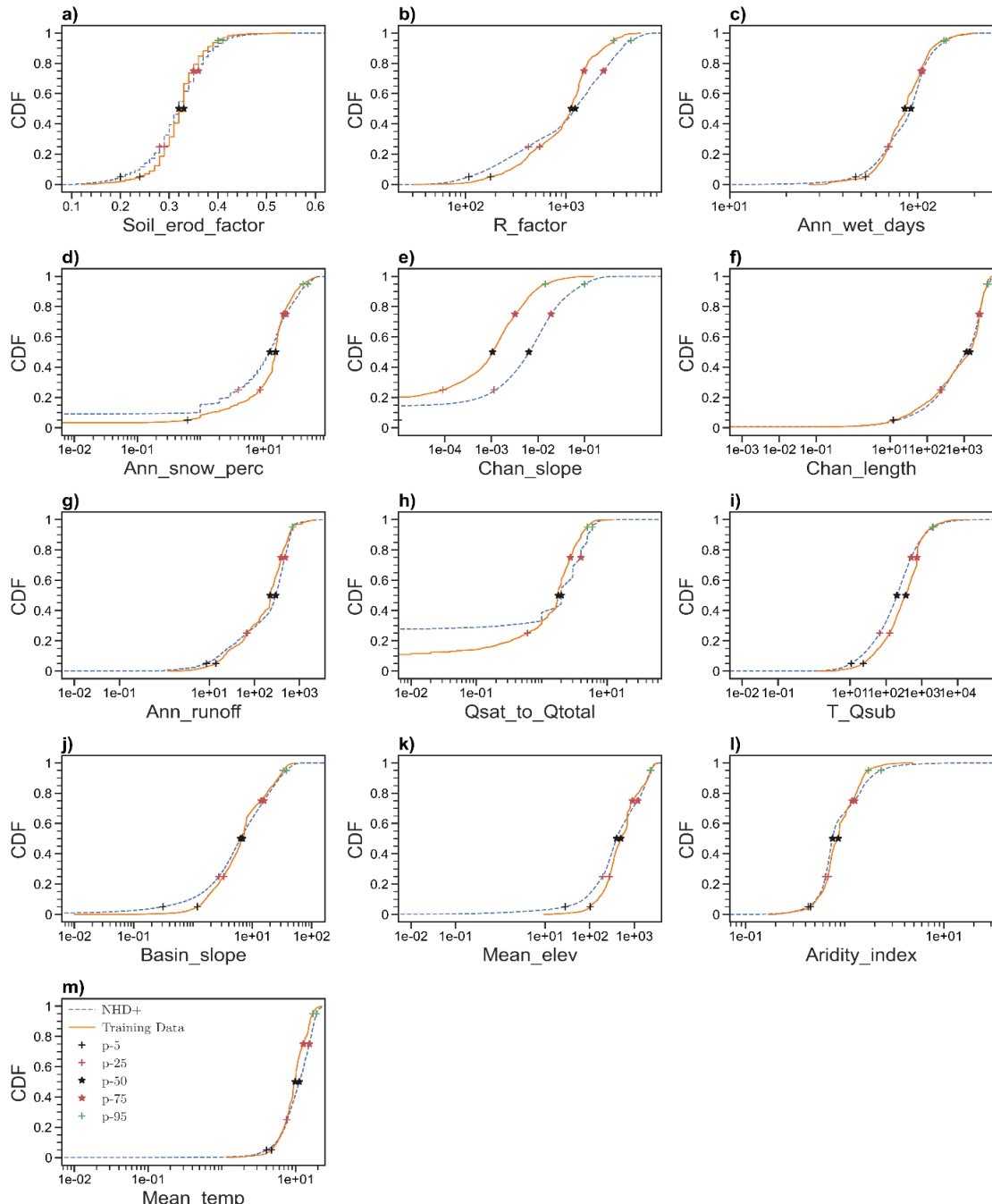


**Figure 5: Comparison of the cumulative distribution function (CDF) of 13 features between training data and all flowlines (i.e., NHDPlus).**

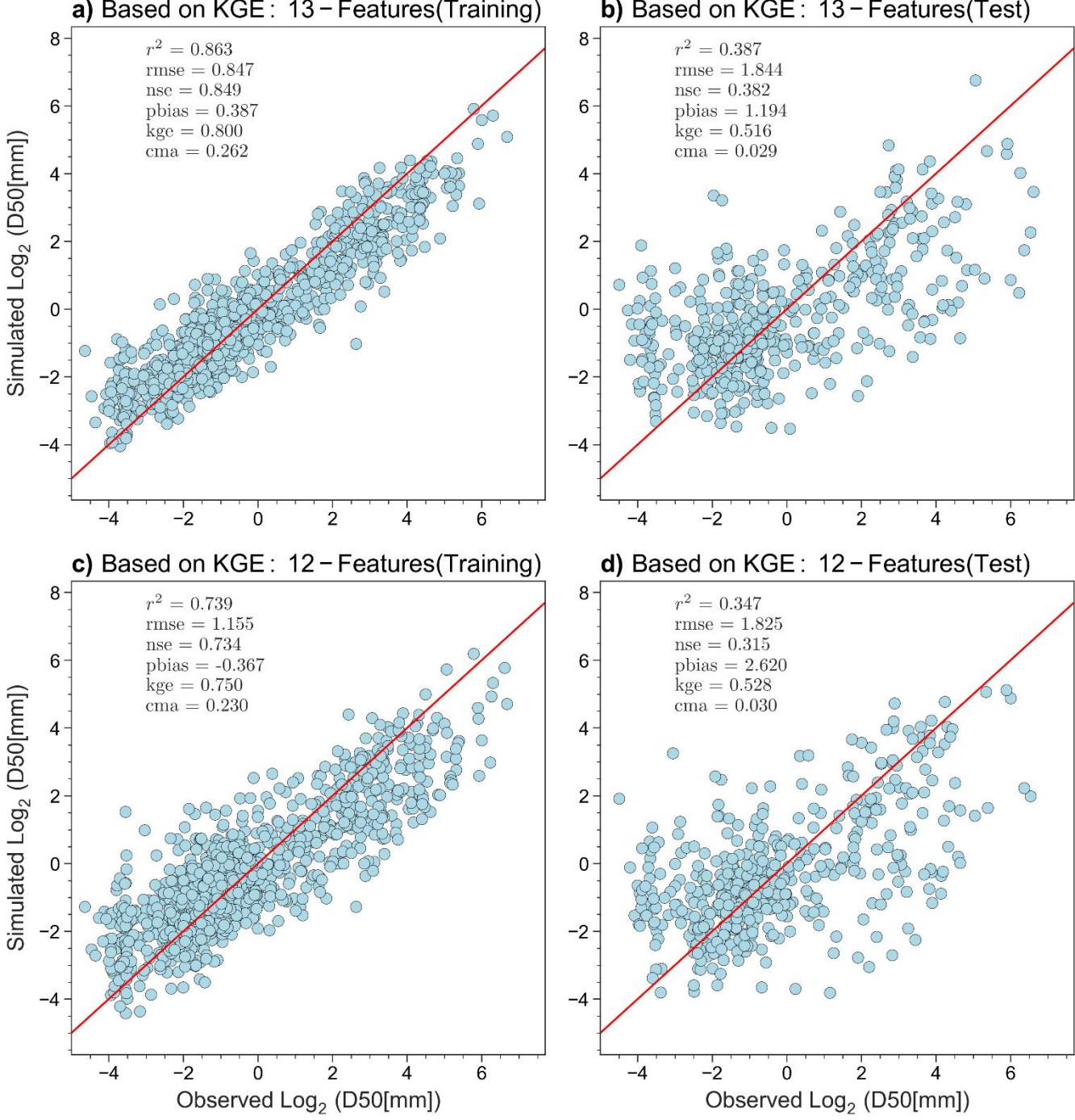

**Figure 6: XGBoost model performance with the training (left) and testing (right) data sets. Comparison of model performances using 13 features (a, b) and 12 features (c,d; after removing "Channel_Mean_Slope").**


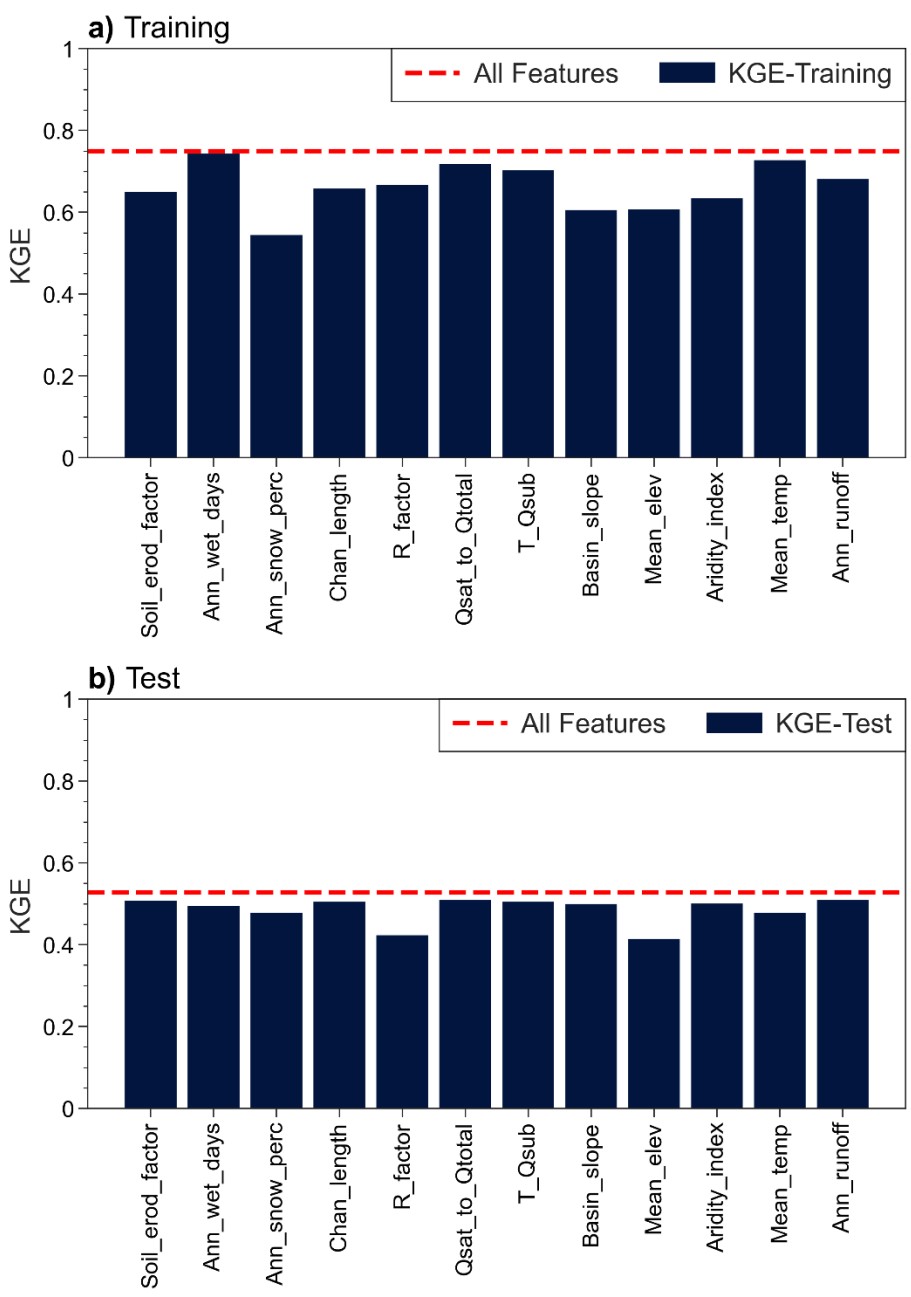

**Figure 7. Sensitivity of the XGBoost model to the selected features. The result shown in blue bars are obtained by dropping the corresponding labelled feature from the 13 selected features. The dashed red line represents the model performance with all variables.**


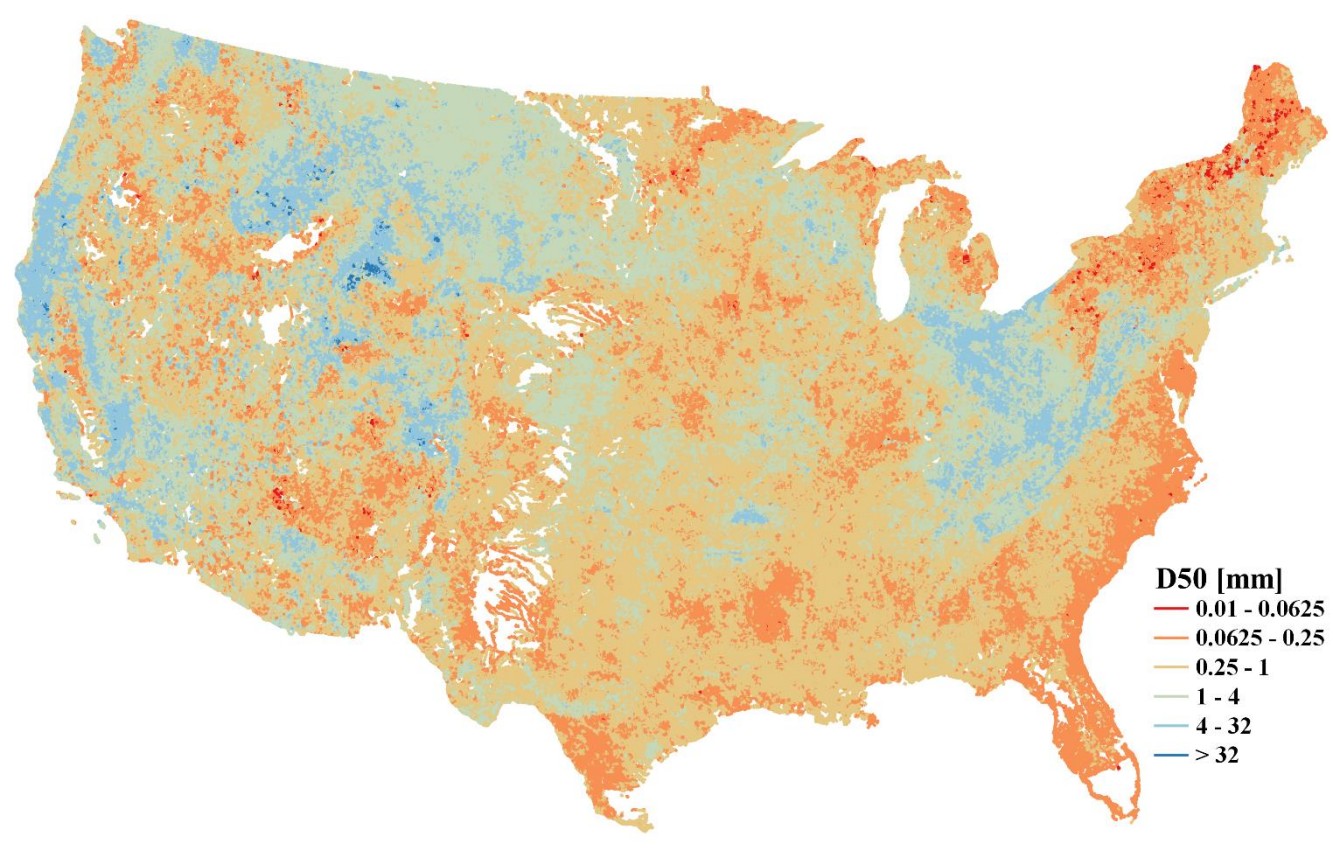

**Figure 8. Predicted D50 in ~2.7 million flowlines across the contiguous U.S. using the XGBoost model.**