# Peer review of "Median bed-material sediment particle size across rivers in the contiguous U.S."

_Earth System Science Data, 2021_

## Referee Comment (RC1)

Review of: **Median bed-material sediment particle size across rivers in the contiguous U.S.**
by *Guta Wakbulcho Abeshu, Hong-Yi Li, Zhenduo Zhu, Zeli Tan, and L. Ruby Leung*

The authors have attempted to solve an important problem of increasing resolution of available sediment D50 in contiguous USA. The authors should be lauded for attempting to tackle this problem using a unique approach of predicting D50 for regions that lacked measurements using a Gradient-Boosting based machine learning method. The paper is well written, though the presentation can be improved. The approach taken by the authors is unique, though I don't think the resultant synthetic dataset generated meets the standards of the measured datasets usually published in this journal. I say this because, it is well known that a ML model's predictive capability is constrained by the range of values present within the training data-set. Even if a trained model shows decent prediction capability for the testing data-set, there is no guarantee that the model will be able to predict correctly for cases that have input parameter values beyond the range of the training data. When the authors synthetically generated the data for the whole US using a model trained using (extremely sparse) 2577 spatial locations, they didn't mention about the range of values for the input parameters in the 26000 flowlines, which were later used to generate the synthetic dataset. I would recommend the authors to submit this paper to a different journal, e.g. WRR, Advances in Water Resources, Geoscientific Model Development, JGRE, etc., and increase the discussion about why the 13 parameters that SHAP value indicate to be the ones most responsible for determining the D50 of sediment at a stream location.

As the above statement about the dataset suitability for the journal could be deemed subjective, I leave the final decision on suitability of the synthetically generated dataset for ESSD to the Editors. Going ahead, I will only comment on the technical and presentation issues of the paper.

1) In line 35 and later, the authors cite "Garcia, 1975". There is not citation in the references to match it.

2) Please re-write line 38 to make is clearer.

3) In line 61, the authors talk about how ML based approaches can allow establishment of successful predictive models without sufficient process-based knowledge. This has proven to be true in different fields; though in others, utilizing ML without a process based understanding has also led to erroneous models that lack generalizability. Often times the difference between success and failure of a ML model is based on the amount of data available for training the model. The authors in the current study have attempted to develop a generalized model for predicting sediment D50 in USA, based on different channel and catchment properties. It is hard to fathom that this could be achieved based on a dataset with only about 2600 data points, without any prior input about the processes involved.

4) In line 72, the authors mention that the dataset has some points with only a single sediment size value. The authors could try to see if they can utilize this extra data, even though D50 calculation is not possible. Maybe the data can be used for further validation of the model.

5) Please include more information in the caption of figure 1 a, e.g. things like how many locations are actually shown on the map. Also, data that is shown as histograms (1 b,c), could/should be represented spatially on the map. This would provide the readers additional information about the spatial variation of different aspects of the data.

6) In all the locations that has multiple values of D50 reported in time, what is the variability in the D50 value over time ? Even though the timescale across which the D50 data was collected is smaller than geomorphological timescales, it is important to check for the variability in order to be sure that the data was collected at stable stream-reaches. Also, what is the scientific basis for calculating a representative D50 by taking a mean ?

7) In line 109, the authors mention that if there are multiple sediment sampling locations for a flowline, they assigned a simple average to come up with the representative D50 for the flowline. This approach is simplistic, as this will work if all the sampling points on a flowline are equidistant. The authors should devise a method that accounts for the relative spatial location of each sampling location, else the representative D50 will be inaccurate.

8) Starting at line 115, the authors mention two studies, specifically Chen and Guestrin (2016) and Zheng et al. (2019), to argue that the ML method they have used is appropriate for the current study. It should be pointed out that even though the XGBoost method that the aforementioned studies used performed admirably, the model developed in those studies were for specific locations. Chen and Guestrin's used it for snowpack spatial patterns in the Sierra Nevada of California, and Zheng et al. used it for predicting water storage changes of a specific lake Inner Mongolia plateau. On the other hand the authors are trying to develop a general model for the whole of USA. Thus, the suitability of the adopted ML technique is debatable.

9) The use of KGE as the model performance parameter is interesting, especially as the KGE values for the testing dataset is relatively much better than the traditional $R^2$. Though, KGE itself is fraught with issues (Onyutha, 2020). So it would be informative if the authors also provide model performance quantification using Nash-Sutcliffe efficiency, CMA (Onyutha, 2020), etc.

10) Once the possible model input parameters has been reduced to 13 parameters (2 channel and 11 basin characteristics), the figures that show results for them (e.g. Fig. 3) should use names of the parameters that are intuitively understandable, rather than something that one has to look up a table (table 1) to recollect. So, please redo the figures.

11) The authors through this exercise of statistically trying to find the most relevant parameters are onto something very interesting and informative. Though, the study isn't complete without a detailed discussion about why or how the parameters that that the model zeros onto are physically connected to the process of sediment D50 formation. Doing this, the reader will have more confidence on the model's predictions and will be a step towards generalization.

12) In line 215, the authors mention that despite lack of any obvious one-on-one correlation between the 13 model input parameters and the D50, they believe the XGBoost model will be able to capture the "high-order interactions" among the input parameters. The authors do not provide any proof to indicate the accuracy of this statement. KGE > 0.5 for the testing dataset is encouraging, though on the other hand dismal $R^2$ (< 0.38) clearly indicate the large amount of dispersion in the model prediction. Thus there is no indication that the model has been able to accurately capture the general trends and processes that decide sediment D50 at a stream-reach. Thus, using this model to synthetically generate possible D50 values is USA, which can then be used model large-scale hydraulic and geomorphological processes is fraught with issues.

13) The authors suggest that the predicted D50 values can be used for producing a map of Manning's roughness coefficient for different streams and reaches in USA. This is hydraulically incorrect. Yes, there are certain stream reaches where D50 is a good indicator of the Manning's roughness coefficient, on the other hand there are many different scenarios under which this will fail. For example, if a stream has vegetation within its flood-plane, the Manning's roughness coefficient will be substantially higher than what the D50 of the channel will predict. Thus, the authors should either remove any mention of Manning's roughness coefficient calculation from the D50 map, or mention the circumstances under which the prediction of Manning's roughness coefficient will be inaccurate.

---

## Referee Comment (RC3)

**Comment on essd-2021-201**

The authors of this paper use machine learning techniques to calculate the median sediment particle size (D50) in U.S. streams. The scarcity of in situ measurements and continuous regional maps make this a worthwhile challenge. A total of 2577 D50 measurements and 76 predictive attributes were used to train a machine learning model and subsequently generate a D50 map for the contiguous U.S. The machine learning model used is a Gradient Boosting variant called XGBoost, its hyperparameters were optimised using the Optuna framework. The model is further improved by trimming the input features through iterative calculation of their feature importance scores. While the main contribution of this work seems to be the resulting National D50 map, I consider the clearly documented ML approach along with a couple of insightful comments on the use of said algorithms to be at least as valuable.

The article is well written and organised and for the most part seems methodically sound, at least from my machine learning point fo view.
My first mayor remark concerns the actual usefulness of the final data product when taking the model performance into consideration: although the KGE might be an established performance metric in the field of hydrology, the testing R2 metric does not point to great predictive accuracy. Ultimately, the model usefulness should be assessed by experts in the field of hydrology (which I am not) and an extended discussion on different performance metrics (including some that facilitate  physical interpretability like RMSE) would help with this assessment.
My second mayor remark regards a possible sample bias and echoes that of the second reviewer albeit from a data focused point of view. The large disparity in counts shown in Fig. 1d) as well as the fact that over 10% of the samples were measured at the same source (USACE Mississippi River main stem) make the question of data representativeness an important one. Along the histogram suggested by the second reviewer, I would suggest a dedicated discussion on how XGBoost handles skewed datasets and their impact on prediction performance.

Beyond this two points, I think the articles makes a good use of the existing data and a well informed use of machine learning to produce a new data product, and I would like to see this research published.

Further minor comments:

Line 85: Is the averaging of samples over time the best way to handle multiple values?

Lines 122-123: This sentence is problematic: while gradient boosting does descend a gradient in some way, it does not make use of *the* Gradient Descent algorithm (Curry, Haskell B., 1944) most machine learning users associate with the concept. Might be worth reformulating or clarifying.

Lines 170-184: A flowchart would be a welcome addition and a good way convey this information at glance.

Lines 187-193: 4.1 would fit better in the Data section.

Lines 208-209: What is the mean annual flow velocity SHAP value and how does it compare to other predictions? Were there any other interesting predictive variables eliminated? A table analogous to Table 1 before feature selection might be useful for this discussion.

Line 235: A title like "Model Sensitivity Analysis" might be more suitable.

Line 285: Could also move to the Data section.

Figure 1a): Seems unnecessary, conveys very little information and could be better described in words (sentence in lines 188-189) or included in Figure with distinct markers.

Figure 1b): X-axis seems half empty and smaller counts are unreadable. Consider reformatting.

Figure 3: Consider flipping the X- or Y- axis order, so as to have a more natural "1.00 diagonal".

---

## Author Comment (AC1)

November 9, 2021

MS. Ref. No. essd-2021-201

"Median bed-material sediment particle size across rivers in the contiguous U.S."

**Dear editor and reviewers,**

The authors would like to thank the reviewers for your time and constructive comments. We are glad that all three reviewers seem to agree about the importance and challenge of the study. The main concern across all reviewers is how well the ranges of input features of training data represent those of the nationwide data, which we agree is very important to look into. To address that, first, we performed a careful, statistical comparison of the 13 input features between the training data and the nationwide data, finding that for 12 out of 13 features the training data are adequately representative of the nationwide data, except for "*CHANNEL\_MEAN\_SLOPE*". Then we removed "*CHANNEL\_MEAN\_SLOPE*" and used the remaining 12 input features to train a new model. It turned out that the new model performance was similar to the previous one in both the training and testing phases. Therefore, we decided to use the new model with the 12 input features and regenerated our results correspondingly. We have also included here some additional analyses or figures to address the other major comments from the reviewers.

Overall, we believe that we have successfully addressed the reviewers' major comments. We have also responded to each of the minor comments. Our point-to-point responses are listed below, where our responses are in blue color and the reviewers' comments are in black color.

**Reviewer #1:**

**https://essd.copernicus.org/preprints/essd-2021-201#RC1**

The authors have attempted to solve an important problem of increasing resolution of available sediment D50 in contiguous USA. The authors should be lauded for attempting to tackle this problem using a unique approach of predicting D50 for regions that lacked measurements using a Gradient-Boosting based machine learning method. The paper is well written, though the presentation can be improved. The approach taken by the authors is unique, though I don't think the resultant synthetic dataset generated meets the standards of the measured datasets usually published in this journal. I say this because, it is well known that a ML model's predictive

capability is constrained by the range of values present within the training data-set. Even if a trained model shows decent prediction capability for the testing data-set, there is no guarantee that the model will be able to predict correctly for cases that have input parameter values beyond the range of the training data. When the authors synthetically generated the data for the whole US using a model trained using (extremely sparse) 2577 spatial locations, they didn't mention about the range of values for the input parameters in the 26000 flowlines, which were later used to generate the synthetic dataset. I would recommend the authors to submit this paper to a different journal, e.g. WRR, Advances in Water Resources, Geoscientific Model Development, JGRE, etc., and increase the discussion about why the 13 parameters that SHAP value indicate to be the ones most responsible for determining the D50 of sediment at a stream location. As the above statement about the dataset suitability for the journal could be deemed subjective, I leave the final decision on suitability of the synthetically generated dataset for ESSD to the Editors. Going ahead, I will only comment on the technical and presentation issues of the paper. Please find detailed comments in the attached document.

Response: We thank the reviewer for the laud and comments.

We agree that "a ML model's predictive capability is constrained by the range of values present within the training data-set". First, per the reviewer's suggestion, we have compared the ranges of the 13 input parameters between the training data and the nationwide data, as shown in Table R1. In Table R1, for each of the 13 features, we used the 2.5th and 97.5th percentiles to represent the lower and higher ends of ranges in the training data. We did not directly use the absolute min/max values to avoid the impacts of outliers. We then calculated the percentage of the nationwide data that is below the 2.5th percentile of the training data, and considered a percentage value no more than 10% indicates a good match of lower ends between the training and nationwide data. Similarly, we calculated the percentage of the nationwide data that is above the 97.5th percentile of the training data, and considered a percentage value no more than 10% indicates a good match of upper ends between the training and nationwide data. Taking together, for any feature, if more than 80% of the nationwide data are located within the 2.5th and 97.5th percentiles of the training data, we consider that the training data is sufficiently representative of the nationwide data for this specific feature. As shown in Table R1, for most features the percentages of the nationwide data that are beyond the lower or higher ends of the training data are no more than 10%, except for channel slope, i.e., "CHANNEL\_MEAN\_SLOPE". Therefore, for 12 out of 13 features, the training data are representative of nationwide data.

Table R1 also lists the relative difference in the 25th, 50th and 75th percentiles between the training and nationwide data. For instance, for the 25th percentile, we calculated the relative difference as the ratio of the difference between the 25th percentile of the training data and that of the nationwide data over the average between the 25th percentile of the training data and that

of the nationwide data. This relative difference is less than 0.5 for most of the features, again except for "*CHANNEL\_MEAN\_SLOPE*", further suggesting that the training data's good representativeness of the nationwide data. For better visual illustration, Figure R1 below shows the box plots. Figure R2 shows the cumulative distribution functions (CDFs) and corresponding 5th-, 25th-, 50th-, 75th-, and 95th-percentiles. In addition, we would like to point out that the 1691 sampling stations we used to train and test our model are located across the whole contiguous U.S., hence geographically representative. Therefore, we conclude that the training data are representative enough of all the flowlines nationwide in terms of the 12 input parameters, except for "*CHANNEL\_MEAN\_SLOPE*".

**Table R1: Comparison of the ranges and percentiles of 13 input features between the training and nationwide datasets.**

|                 | Percent of nationwide |           | Relative difference in percentiles between |      |      |
|-----------------|-----------------------|-----------|--------------------------------------------|------|------|
| Attributes      | data                  |           |                                            |      |      |
|                 |                       |           | the training and                           |      |      |
|                 |                       |           | nationwide data                            |      |      |
|                 | below                 | above     | 25th                                       | 50th | 75th |
|                 | 2.5th of              | 97.5th of |                                            |      |      |
|                 | training              | training  |                                            |      |      |
|                 | data                  | data      |                                            |      |      |
| TOT_KFACT       | 5.1                   | 3.4       | 0.07                                       | 0.03 | 0.03 |
| TOT_RFACT       | 6.9                   | 10.2      | 0.22                                       | 0.09 | 0.44 |
| TOT_WDANN       | 2.8                   | 3.1       | 0.01                                       | 0.07 | 0.03 |
| TOT_PRSNOW      | 0                     | 3.9       | 0.71                                       | 0.20 | 0.12 |
| CHANNEL_MEAN_SL | 0                     | 19.9      | 1.72                                       | 1.44 | 1.43 |
| OPE             |                       |           |                                            |      |      |
| PATHLENGTH      | 2.3                   | 5.2       | 0.07                                       | 0.21 | 0.04 |
| TOT_RUN7100     | 4.3                   | 1.6       | 0                                          | 0.28 | 0.23 |
| TOT_SATOF       | 0                     | 7.1       | 2                                          | 0.13 | 0.38 |
| TOT_CONTACT     | 6.2                   | 3.1       | 0.68                                       | 0.59 | 0.37 |
| TOT_BASIN_SLOPE | 9.3                   | 4.8       | 0.17                                       | 0.04 | 0.10 |
| TOT_ELEV_MEAN   | 9.9                   | 2.5       | 0.35                                       | 0.27 | 0.22 |
| AI              | 2.6                   | 6.5       | 0.07                                       | 0.14 | 0.03 |
| TOT_WBM_TAV     | 4.5                   | 8.6       | 0.02                                       | 0.12 | 0.18 |

Figure R1: Box plots for the comparison of the ranges of 13 parameters between training data and all flowlines (i.e., NHD+).

---

## Author Response (AR2)

December 18, 2021

MS. Ref. No. essd-2021-201

"Median bed-material sediment particle size across rivers in the contiguous U.S."

**Topic Editor:**

Dear Authors

We received three reviews of your manuscript. The reviewers agree that this data is useful in principle, but ask for a few points to be addressed in more detail to describe the quality and usefulness of the resulting data product.

Kind regards

Jens Klump

Dear editor and reviewers,

The authors would like to thank the editor and three reviewers for your time and constructive comments. We are glad that all three reviewers seem to agree about the importance and challenge of the study. The main concern across all reviewers is how well the ranges of input features of training data represent those of the nationwide data, which we agree is very important to look into. To address that, first, we performed a careful, statistical comparison of the 13 input features between the training data and the nationwide data, finding that for 12 out of 13 features the training data are adequately representative of the nationwide data, except for "*CHANNEL_MEAN_SLOPE*". Then we removed "*CHANNEL_MEAN_SLOPE*" and used the remaining 12 input features to train a new model. It turned out that the new model performance was similar to the previous one in both the training and testing phases. Therefore, we decided to use the new model with the 12 input features and regenerated our results correspondingly. We have also included here some additional analyses or figures to address the other major comments from the reviewers.

Overall, we believe that we have successfully addressed the reviewers' major comments. We have also responded to each of the minor comments. Our point-to-point responses are listed below, where our responses are in blue color and the reviewers' comments are in black color.

**Please note that the line numbers we provide in the below are corresponding to the revised manuscript with the changes tracked, not the clean version.**

**Reviewer #1:**

https://essd.copernicus.org/preprints/essd-2021-201#RC1

The authors have attempted to solve an important problem of increasing resolution of available sediment D50 in contiguous USA. The authors should be lauded for attempting to tackle this problem using a unique approach of predicting D50 for regions that lacked measurements using a Gradient-Boosting based machine learning method. The paper is well written, though the presentation can be improved. The approach taken by the authors is unique, though I don't think the resultant synthetic dataset generated meets the standards of the measured datasets usually published in this journal. I say this because, it is well known that a ML model's predictive capability is constrained by the range of values present within the training data-set. Even if a trained model shows decent prediction capability for the testing data-set, there is no guarantee that the model will be able to predict correctly for cases that have input parameter values beyond the range of the training data. When the authors synthetically generated the data for the whole US using a model trained using (extremely sparse) 2577 spatial locations, they didn't mention about the range of values for the input parameters in the 26000 flowlines, which were later used to generate the synthetic dataset. I would recommend the authors to submit this paper to a different journal, e.g. WRR, Advances in Water Resources, Geoscientific Model Development, JGRE, etc., and increase the discussion about why the 13 parameters that SHAP value indicate to be the ones most responsible for determining the D50 of sediment at a stream location. As the above statement about the dataset suitability for the journal could be deemed subjective, I leave the final decision on suitability of the synthetically generated dataset for ESSD to the Editors. Going ahead, I will only comment on the technical and presentation issues of the paper. Please find detailed comments in the attached document.

Response: We thank the reviewer for the laud and comments.

We agree that "a ML model's predictive capability is constrained by the range of values present within the training data-set". First, per the reviewer's suggestion, we have compared the ranges of the 13 input parameters between the training data and the nationwide data, as shown in Table 2. In Table 2, for each of the 13 features, we used the 2.5th and 97.5th percentiles to represent the lower and higher ends of ranges in the training data. We did not directly use the absolute min/max values to avoid the impacts of outliers. We then calculated the percentage of the nationwide data that is below the 2.5th percentile of the training data, and considered a percentage value no more than 10% indicates a good match of lower ends between the training and nationwide data. Similarly, we calculated the percentage of the nationwide data that is above

the 97.5th percentile of the training data, and considered a percentage value no more than 10% indicates a good match of upper ends between the training and nationwide data. Taking together, for any feature, if more than 80% of the nationwide data are located within the 2.5th and 97.5th percentiles of the training data, we consider that the training data is sufficiently representative of the nationwide data for this specific feature. As shown in Table 2, for most features the percentages of the nationwide data that are beyond the lower or higher ends of the training data are no more than 10%, except for channel slope, i.e., "*CHANNEL_MEAN_SLOPE*". Therefore, for 12 out of 13 features, the training data are representative of nationwide data.

Table 2 also lists the relative difference in the 25th, 50th and 75th percentiles between the training and nationwide data. For instance, for the 25th percentile, we calculated the relative difference as the ratio of the difference between the 25th percentile of the training data and that of the nationwide data over the average between the 25th percentile of the training data and that of the nationwide data. This relative difference is less than 0.5 for most of the features, again except for "*CHANNEL_MEAN_SLOPE*", further suggesting that the training data's good representativeness of the nationwide data. For better visual illustration, Figure 5 shows the cumulative distribution functions (CDFs) and corresponding 5th-, 25th-, 50th-, 75th-, and 95th-percentiles, and Figure S6 shows the box plots. In addition, we would like to point out that the 1691 sampling stations we used to train and test our model are located across the whole contiguous U.S., hence geographically representative. Therefore, we conclude that the training data are representative enough of all the flowlines nationwide in terms of the 12 input parameters, except for "*CHANNEL_MEAN_SLOPE*".

We removed "*CHANNEL_MEAN_SLOPE*" and used the remaining 12 parameters to develop a new model, following the same model training and testing procedures as before. Figure R3 below shows the comparison of model performances between the previous and new models. The model performance metrics are very similar. Actually, $R^2$ became slightly better in both training (0.834 vs. 0.830) and testing (0.405 vs. 0.367), while KGE became slightly worse in training (0.775 vs. 0.794) but better in testing (0.527 vs. 0.513). The slight decrease of KGE in training data is reasonable since the model hyperparameter tuning was based on the objective of maximizing KGE and losing one parameter will slightly reduce the space of parameter tuning. Nevertheless, now the KGE value in the testing phase is closer to that in the training phase.

In summary, we think we have successfully addressed the reviewer's primary concern by performing the statistical comparison between the training and nationwide data, and rerunning the ML model using the further refined selection of input parameters. In the revised manuscript, we plan to add those comparisons and discussion, which we believe will substantially elevate our study.

Regarding the reviewer's comment on the suitability of our manuscript for ESSD, we did carefully consider various journals and then decided that ESSD was most suitable for our study.

Most importantly, there have been articles published at ESSD that are similar to ours, i.e., generating a dataset at a regional or global scale from a relatively small number of local/point observations. Here're two very recent examples:
https://essd.copernicus.org/articles/13/4881/2021/;
https://essd.copernicus.org/articles/13/3453/2021/essd-13-3453-2021.html.

Action: We added lines 144-145, 195-204, and 257-278.

1) In line 35 and later, the authors cite "Garcia, 1975". There is not citation in the references to match it.

Response: It should be Garcia, 2008.

Action: We fixed it in the revised manuscript, lines 36 and 38.

2) Please re-write line 38 to make it clearer.

Action: We re-wrote lines 39-41.

3) In line 61, the authors talk about how ML based approaches can allow establishment of successful predictive models without sufficient process-based knowledge. This has proven to be true in different fields; though in others, utilizing ML without a process-based understanding has also led to erroneous models that lack generalizability. Oftentimes the difference between success and failure of a ML model is based on the amount of data available for training the model. The authors in the current study have attempted to develop a generalized model for predicting sediment D50 in the USA, based on different channel and catchment properties. It is hard to fathom that this could be achieved based on a dataset with only about 2600 data points, without any prior input about the processes involved.

Response: For developing a generalized model for predicting D50, as discussed above, it is critical whether or not the ranges of input parameters of the training data are representative for those of the river reaches across the nation. With Table 2 and Figures 5, S6 and 6, we believe the training data used in our revised ML model, although not a great amount, represent reasonably well the nationwide data. Also, the testing data are effective for ML approaches in order to check how well predictive the ML model is. Lastly, we actually did invest quite some efforts in the traditional, regression-type of methods. We first examined the causal relationships between D50 and each variable, for example, stream order, channel slope, mean annual flow, mean flow velocity, channel sinuosity, channel hydraulic geometries etc. However, there has been too much scattering in each of these relationships (hence uncertainty; see Figure S5), preventing the

subsequent regression analysis. Therefore, we eventually decided to rely on the ML methods due to lack of explicit understanding of physics. Our rationale is that, the traditional regression or dimensionless analysis techniques may not work in generating a large-scale spatial map of D50, and ML offers a great opportunity for us to move forward, i.e., as pointed out by the 2nd reviewer, making some useful progress instead of keeping waiting.

Action: We added Figure S5 in Supplement Material and modified lines 253-256.

4)      In line 72, the authors mention that the dataset has some points with only a single sediment size value. The authors could try to see if they can utilize this extra data, even though D50 calculation is not possible. Maybe the data can be used for further validation of the model.

Response: We have looked into those USGS stations with some sediment particle size measurements but D50 calculation is not possible. There are 1367 USGS stations with incomplete percentiles of bed-material sediment particle data, which can be divided into three groups: 1) 1183 stations have no effective percentiles provided; 2) 147 stations have only percentiles above the 50$^{th}$ percentile; 3) 37 stations have only percentiles below the 50$^{th}$ percentile. We feel that Therefore, we neglect these data in further analysis.

Action: We added above clarification in lines 74-77.

5)      Please include more information in the caption of figure 1a, e.g. things like how many locations are actually shown on the map. Also, data that is shown as histograms (1b,c), could/should be represented spatially on the map. This would provide the readers additional information about the spatial variation of different aspects of the data.

Action: We added the location information in the caption of Figure 1. For the data shown as histograms (Fig. 1b,c), we plotted the data spatially as maps in Supplement Material Figure S1. We also modified lines 83-86.

6)      In all the locations that have multiple values of D50 reported in time, what is the variability in the D50 value over time? Even though the timescale across which the D50 data was collected is smaller than geomorphological timescales, it is important to check for the variability in order to be sure that the data was collected at stable stream-reaches. Also, what is the scientific basis for calculating a representative D50 by taking a mean?

Response: We calculated the coefficient of variation (CV) for the 760 stations that have at least 5 samples over time. For the rest of stations, the number of samples are too small for meaningful calculation of CV. Figure S2 in Supplement Material shows that for most of these 760 stations the CV values range between 0.3 and 1.2 with the median of approximately 0.6. The small CV values indicate the good stability of D50 (at the same location) over time. We took the mean as a

representative D50 to simply account for possible uncertainties in sampling and measurement. Although the sampling and measurement procedures were carefully designed (see Edwards and Glysson methods document https://pubs.usgs.gov/twri/twri3-c2/, as suggested by reviewer #2), it is practically impossible to avoid uncertainties in such sampling and measurement procedures. Thus, we believe a representative D50 can be better estimated by taking a mean.

Action: We added above clarification in lines 86-89.

7)     In line 109, the authors mention that if there are multiple sediment sampling locations for a flowline, they assigned a simple average to come up with the representative D50 for the flowline. This approach is simplistic, as this will work if all the sampling points on a flowline are equidistant. The authors should devise a method that accounts for the relative spatial location of each sampling location, else the representative D50 will be inaccurate.

Response: We agree with the reviewer that D50 probably changes spatially within a flowline. We did the simple average for two reasons: 1) Only a very small number of flowlines have more than one sampling locations. To be exact, there are only 12 flowlines with 2 sampling locations and another 2 flowlines with 3 sampling locations. 2) The mean length of the 14 flowlines is 6.63km. In such a length, only two or three sampling locations cannot capture the spatial variability in a meaningful way. Therefore, we simply calculated the average without making further assumptions.

Action: We added above clarification in lines 94-97.

8)     Starting at line 115, the authors mention two studies, specifically Chen and Guestrin (2016) and Zheng et al. (2019), to argue that the ML method they have used is appropriate for the current study. It should be pointed out that even though the XGBoost method that the aforementioned studies used performed admirably, the model developed in those studies were for specific locations. Chen and Guestrin's used it for snowpack spatial patterns in the Sierra Nevada of California, and Zheng et al. used it for predicting water storage changes of a specific lake Inner Mongolia plateau. On the other hand the authors are trying to develop a general model for the whole of the USA. Thus, the suitability of the adopted ML technique is debatable.

Response: We agree that the suitability of XGBoost or any ML technique cannot be guaranteed based on the success of other studies on different problems. As discussed above, we have ensured that the training data we used are representative for the whole U.S. The sampling locations of the training data spread over most of the U.S., hence geographically representative as well. Lastly, a practical way to check whether or not it is suitable is by applying the method and evaluating its performance with testing data. In our results the model performance in the testing phase is sufficiently close to that in the training phase, further indicating that our ML model is suitable for the other places in the U.S.

9)      The use of KGE as the model performance parameter is interesting, especially as the KGE values for the testing dataset is relatively much better than the traditional $R^2$. Though, KGE itself is fraught with issues (Onyutha, 2020). So it would be informative if the authors also provide model performance quantification using Nash-Sutcliffe efficiency, CMA (Onyutha, 2020), etc.

Response: In fact, we have used KGE, NSE and $R^2$ as the model performance parameters (hence objective functions). Based on a visual check of the patterns we obtained (see Figure S7), we feel that using KGE gives better patterns, i.e., the dots are more aligned with the 1:1 line as indicated by the percentage of bias (PBIAS).

Action: We added Figure S7 in Supplement Material and lines 307-310. Per the reviewers' suggestion, we also added more metrics in Figure 6, including CMA.

10)     Once the possible model input parameters has been reduced to 13 parameters (2 channel and 11 basin characteristics), the figures that show results for them (e.g. Fig. 3) should use names of the parameters that are intuitively understandable, rather than something that one has to look up a table (table 1) to recollect. So, please redo the figures.

Action: We used new names that are more intuitively understandable in Tables 1 and 2, Figures 4, 5, and 7.

11)     The authors through this exercise of statistically trying to find the most relevant parameters are onto something very interesting and informative. Though, the study isn't complete without a detailed discussion about why or how the parameters that the model zeros onto are physically connected to the process of sediment D50 formation. Doing this, the reader will have more confidence in the model's predictions and will be a step towards generalization.

Response: It would be indeed interesting to explore and reveal how these parameters are physically connected to the processes of D50 formation. Unfortunately, we believe that such a study (maybe even a few studies are needed) is beyond the scope of our current study because it would require 1) a highly-integrated, process-based model that considers at least sediment erosion, deposition and transport processes in both hillslopes and channels, and 2) well-designed numerical experiments to isolate the dominant processes and controlling factors. The relations between the input parameters (e.g. watershed characteristics) and D50 are too complex to be revealed with traditional linear regression or dimensionless analysis methods (In fact, we did spend some time on it as well but have not been successful). Therefore, we decided to use XGBoost due to its satisfactory performance without invoking the related physics explicitly. However, an unpleasant compromise comes along with the XGBoost model is its limitation of explainability (and this is true for other machine learning models).

Action: We added above clarification in lines 141-142, 284-287, and 341-343.

12)     In line 215, the authors mention that despite lack of any obvious one-on-one correlation between the 13 model input parameters and the D50, they believe the XGBoost model will be able to capture the "high-order interactions" among the input parameters. The authors do not provide any proof to indicate the accuracy of this statement. KGE > 0.5 for the testing dataset is encouraging, though on the other hand dismal $R^2$ ($< 0.38$) clearly indicates the large amount of dispersion in the model prediction. Thus there is no indication that the model has been able to accurately capture the general trends and processes that decide sediment D50 at a stream-reach. Thus, using this model to synthetically generate possible D50 values is USA, which can then be used to model large-scale hydraulic and geomorphological processes is fraught with issues.

Response: We thank the reviewer for the interesting discussion. In deriving a dataset (with spatial or temporal inter- or extrapolation), one traditional way is to have sufficient process-based understanding first and then derive the data based on the understanding, which is the path that the reviewer is suggesting, and unfortunately, has not been working in this specific issue of deriving a large-scale D50 dataset (not for lack of trying), otherwise this would have not been a long-standing challenge. In our 12 selected parameters, only 1 is directly related to the channel processes. The remaining 11 are all land parameters and their mechanistic connections with D50 are rather mysterious at this stage. This fact partially explains why the traditional way has not been working (certainly not for lack of trying), and we have to rely on the ML methods. That said, what we are presenting in this study could be used as empirical evidence on the likely causal, yet highly complicated relationships between D50 and the land parameters, and hopefully inspire future studies to shed light on the underlying mechanisms. We have tested the usage of the new D50 dataset within a large-scale suspended modeling framework (see https://hess.copernicus.org/preprints/hess-2021-491/), and our successful model validation against the USGS observed suspended sediment load over multiple stations suggests the good value of such a D50 dataset.

Action: We added description about the performance of a large-scale suspended sediment modeling study that used the data from this paper in lines 355-357.

13)     The authors suggest that the predicted D50 values can be used for producing a map of Manning's roughness coefficient for different streams and reaches in the USA. This is hydraulically incorrect. Yes, there are certain stream reaches where D50 is a good indicator of the Manning's roughness coefficient, on the other hand there are many different scenarios under which this will fail. For example, if a stream has vegetation within its flood-plane, the Manning's roughness coefficient will be substantially higher than what the D50 of the channel will predict. Thus, the authors should either remove any mention of Manning's roughness coefficient calculation from the D50 map, or mention the circumstances under which the prediction of Manning's roughness coefficient will be inaccurate.

Response: We agree predicting Manning's roughness coefficient using D50 will be problematic so we decided to remove this.

Action: We deleted lines 362-367.

**Reviewer #2:**

https://essd.copernicus.org/preprints/essd-2021-201#RC2

The authors have taken up an important and interesting problem of estimating D50 in streams across the United States, when limited data exists to make these estimates. I think the data set and analysis is publishable and will be of use to the scientific community, despite the inherent uncertainties in the estimates in stream systems where little or no training data is available. The data used for the analysis has been collected for many years, and if the community waits until sufficient data is present to train the estimation model more accurately, we will be waiting for a very long time. With this being said, I think the authors need to provide more clarification regarding where their model estimates can be considered stronger and where they should be considered weaker; also some discussion regarding the representativeness of bed sediment data at gauge stations is warranted.

Response: We appreciate the reviewer's positive and constructive comment. We have addressed the specific comments as below.

My biggest concern is that smaller streams may have more limited training data than exists for larger streams, and the D50 estimates for these smaller streams will involve greater uncertainty. This is based on the following: (a) a greater percentage of large streams are gaged by the USGS than small streams; (b) most of the data that does exist has D50 in the sand range – this suggests that smaller, steeper first and second order streams that are more likely to be gravel/cobble bedded are poorly represented in the data set, even though these smaller streams may dominate the total length of streams in the database. To clarify this issue as to whether my concern is founded, it would be helpful for the authors to provide a figure with two histograms for comparison: (1) a histogram that bins the stream database data according to stream size (x-axis) and analyzes total length of stream in each bin (y-axis); (2) a histogram that utilizes the same stream size (x-axis) bins and analyzes the number of D50 data points in each bin (y-axis). The variable chosen to represent stream size will need to be a surrogate such as upstream catchment area of the reach (*totdasqkm*) or stream order (*streamorde*) attributes. If that analysis confirms that my concern is founded, my recommendation will probably be that the analysis should be stratified into a dataset with higher confidence (larger streams) and a separate dataset with lower confidence (smaller streams).

Response: This is indeed a constructive comment. We plotted the histograms as the reviewer suggested (see Figure S3). For stream size, we quantified it using stream order (Fig. S3a,b) and upstream drainage area (Fig. S3c,d), respectively. Interestingly, smaller streams do not dominate total flowline length. Instead, the total flowline length of each stream order generally increases as stream size increases. The number of D50 stations follows a bell distribution except for the

largest stream order or drainage area, which is primarily due to the USACE measurements on the lower Mississippi River (198 sample locations). Therefore, there's no clear indication that the D50 data points are dominated by either larger or smaller streams. Our additional analysis here thus gives more confidence in the way the training data are treated in our study. In the revised manuscript, we will add some discussion on the possible impact of larger/smaller streams. We hope these additional analysis and discussion can satisfactorily address the reviewer's concern.

Action: We added lines 104-111 and Figure S3 in Supplement Material.

Secondly, I think some discussion is warranted regarding what the D50 data at the gauge stations represents. Gage stations are established at cross sections in the stream where flow measurements are convenient and with conditions conducive to high quality flow measurements – the issue of whether the bed sediment composition represents the reach is generally not taken into account when the gage station location is established. A large percentage of gage stations are at bridges. Due to flow constriction at flood stage, bridge sections are more likely to be subject to significant scour and fill – thus, with the bed sediment being more representative of the pool than the riffle or cross-over in a reach. Particularly in gravel-bedded streams, the difference in bed sediment composition between the pool and the riffle can be substantial. This does not invalidate the data; the analysis just needs to be clear that the bed sediment at the gage station may not always be representative of the reach, which will help the user of the data understand its limitations. I would recommend citing the Edwards and Glysson methods document (https://pubs.usgs.gov/twri/twri3-c2/), which probably best characterizes how most of the bed sediment samples were collected and composited at a cross section by the USGS over the years. A short conversation with Molly Wood or Tim Straub of the USGS would also be useful to inform your discussion of these issues.

Action: Good suggestion. We added lines 345-350 to discuss this issue.

**Reviewer #3:**

https://doi.org/10.5194/essd-2021-201-RC3

The authors of this paper use machine learning techniques to calculate the median sediment particle size (D50) in U.S. streams. The scarcity of in situ measurements and continuous regional maps make this a worthwhile challenge. A total of 2577 D50 measurements and 76 predictive attributes were used to train a machine learning model and subsequently generate a D50 map for the contiguous U.S. The machine learning model used is a Gradient Boosting variant called XGBoost, its hyperparameters were optimised using the Optuna framework. The model is further improved by trimming the input features through iterative calculation of their feature importance scores. While the main contribution of this work seems to be the resulting National D50 map, I consider the clearly documented ML approach along with a couple of insightful comments on the use of said algorithms to be at least as valuable. The article is well written and organised and for the most part seems methodically sound, at least from my machine learning point of view.

Response: We appreciate the reviewer's positive and constructive comment. We have addressed the specific comments as below.

My first major remark concerns the actual usefulness of the final data product when taking the model performance into consideration: although the KGE might be an established performance metric in the field of hydrology, the testing R2 metric does not point to great predictive accuracy. Ultimately, the model usefulness should be assessed by experts in the field of hydrology (which I am not) and an extended discussion on different performance metrics (including some that facilitate physical interpretability like RMSE) would help with this assessment.

Response: This concern about $R^2$ and RMSE metrics is similar to reviewer #1's comment #9. In a nutshell, we have used $R^2$ and NSE as the objective functions in addition to KGE, and eventually decided to go with KGE for seemingly better effectiveness. We have also tested the usefulness of our dataset within a large-scale suspended sediment modeling framework (see https://hess.copernicus.org/preprints/hess-2021-491/). For more details, please refer to our responses above to Reviewer #1.

Action: We added Figure S7 in Supplement Material and lines 307-310.

My second major remark regards a possible sample bias and echoes that of the second reviewer albeit from a data focused point of view. The large disparity in counts shown in Fig. 1d) as well as the fact that over 10% of the samples were measured at the same source (USACE Mississippi River main stem) make the question of data representativeness an important one. Along the

histogram suggested by the second reviewer, I would suggest a dedicated discussion on how XGBoost handles skewed datasets and their impact on prediction performance.

Response: The additional analyses based on the Reviewer #2's suggestion show that there is no obvious sampling bias between the larger and smaller streams.

Action: We added discussion on the possible sampling biases as pointed out by Reviewer #2 (e.g., impacts of bridge sections) and echoed by Reviewer #3 in lines 345-350.

Beyond this two points, I think the articles makes a good use of the existing data and a well-informed use of machine learning to produce a new data product, and I would like to see this research published.

Response: We appreciate the reviewer's positive comment.

Further minor comments:

Line 85: Is the averaging of samples over time the best way to handle multiple values?

Response: Averaging samples is based on the fact there's no significant variability in D50 sampled over time for most stations as shown in Figure R5. Please refer to our responses to Reviewer #1's comment #6 for detailed discussion on D50's temporal variability.

Lines 122-123: This sentence is problematic: while gradient boosting does descend a gradient in some way, it does not make use of the Gradient Descent algorithm (Curry, Haskell B., 1944) most machine learning users associate with the concept. Might be worth reformulating or clarifying.

Action: We modified line 149 to clarify this issue.

Lines 170-184: A flowchart would be a welcome addition and a good way convey this information at glance.

Action: We added a flowchart as Figure 3.

Lines 187-193: 4.1 would fit better in the Data section.

Response: We tend to keep this subsection in the Results section because treatment of the training data is an important step and the training data are not the same as those raw data. We

feel keeping it here can help avoid confusion and clearly indicate that the training data (e.g., Figure 2) could be different if choosing different treatments.

Lines 208-209: What is the mean annual flow velocity SHAP value and how does it compare to other predictions? Were there any other interesting predictive variables eliminated? A table analogous to Table 1 before feature selection might be useful for this discussion.

Response: The feature selection using SHAP is an iterative procedure so we actually don't have a single SHAP value for the mean annual flow velocity. We agree with the reviewer that there might be some interesting predictive variables eliminated during our selection procedure due to the low SHAP values. Interestingly, in our final 12 selected predictive variables, 11 are not riverine variables, but hydroclimatological or landscape properties. In fact, we actually did try to examine the individual relationship between D50 and each of some seemingly important, candidate predictive variables, for example, mean annual flow velocity, stream order, channel slope, mean annual flow, channel sinuosity, channel hydraulic geometries etc. However, we could not find any clear pattern or relationship due to too much uncertainty (scattering), preventing the subsequent regression analysis.

Line 235: A title like "Model Sensitivity Analysis" might be more suitable.

Action: Agree. We changed it in line 300.

Line 285: Could also move to the Data section.

Response: This section of "Data Availability" is required by ESSD. Please see the instructions here: https://www.earth-system-science-data.net/submission.html

Figure 1a): Seems unnecessary, conveys very little information and could be better described in words (sentence in lines 188-189) or included in Figure with distinct markers.

Response: The intention of this subfigure was to indicate the locations of sampling data from two sources (USGS and USACE). We have thought about merging this info into Figure 2, i.e., using distinct markers but eventually decided not to do so for two reasons: 1) Some people care about such info, e.g., Reviewer #1; 2) Figures 1 and 2 actually display different geometric features, i.e., the former displays points (USGS stations) and the latter displays lines (flowlines).

Figure 1b): X-axis seems half empty and smaller counts are unreadable. Consider reformatting.

Action: We changed the y-axis to be log-scale.

Figure 3: Consider flipping the X- or Y- axis order, so as to have a more natural "1.00 diagonal"

Action: Done.